# SecureFedYJ: a safe feature Gaussianization protocol for Federated Learning

**Tanguy Marchand**
Owkin Inc., New York, USA.
`tanguy.marchand@owkin.com`

**Boris Muzellec**
Owkin Inc., New York, USA.
`boris.muzellec@owkin.com`

**Constance Beguier**[*]

**Jean Ogier du Terrail**
Owkin Inc., New York, USA.
`jean.du-terrail@owkin.com`

**Mathieu Andreux**
Owkin Inc., New York, USA.
`mathieu.andreux@owkin.com`

## Abstract

The Yeo-Johnson (YJ) transformation is a standard parametrized per-feature unidimensional transformation often used to Gaussianize features in machine learning. In this paper, we investigate the problem of applying the YJ transformation in a cross-silo Federated Learning setting under privacy constraints. For the first time, we prove that the YJ negative log-likelihood is in fact convex, which allows us to optimize it with exponential search. We numerically show that the resulting algorithm is more stable than the state-of-the-art approach based on the Brent minimization method. Building on this simple algorithm and Secure Multiparty Computation routines, we propose SECUREFEDYJ, a federated algorithm that performs a pooled-equivalent YJ transformation without leaking more information than the final fitted parameters do. Quantitative experiments on real data demonstrate that, in addition to being secure, our approach reliably normalizes features across silos as well as if data were pooled, making it a viable approach for safe federated feature Gaussianization.

## 1  Introduction

Federated Learning (FL) [45, 32] is an approach that was recently proposed to train machine learning (ML) models across multiple data holders, or *clients*, without centralizing data points, notably for privacy reasons. While many FL applications have been proposed, two main settings have emerged [23]: cross-device FL, involving a large number of small edge devices, and cross-silo FL, dealing with a smaller number of clients, with larger computational capabilities. Due to the sensitivity and relative local scarcity of medical data, healthcare is a promising application of cross-silo FL [40], e.g. to train a biomedical ML model between different hospitals as if all the datasets were pooled in a central server. In this paper, we focus on the cross-silo setting.

**The constraints of cross-silo FL**  Although cross-silo FL resembles standard distributed learning, it faces at least two important distinct challenges: privacy and heterogeneity. Due to data sensitivity, clients might impose stringent security and privacy constraints on FL collaborations. This arises in *coopetitive* FL projects, where models are jointly trained on industrial competitors' datasets [55], as well as medical FL applications, where conservative data regulations might apply. In this setting, using standard FL algorithms such as FEDAVG [32] might not provide enough privacy guarantees, as privacy attacks such as data reconstruction can be carried out based on the clients' gradients [56, 54].

---

[*]Contribution done while at Owkin, Inc.

36th Conference on Neural Information Processing Systems (NeurIPS 2022).

Various protocols based on Secure Multiparty Computation (SMC) (see Section 2 for more details), such as Secure Aggregation [4], can mitigate this shortcoming by disclosing only the sum of the gradients from all clients to the server, without disclosing each gradient individually.

An additional constraint is that data might present statistical heterogeneity across clients, i.e. the local clients' data distributions may not be identical. In the case of medical applications, such heterogeneity may be caused e.g. by environmental variations or differences in the material that was used for acquisition [43, 47, 2]. While different ways of adapting federated training algorithms have been proposed to automatically tackle heterogeneity [28, 29, 24], these solutions do not address data harmonization and normalization prior to FL training.

**Preprocessing in ML** Data preprocessing is a crucial step in many ML applications, leading to important performance gains. Among others, common preprocessing methods include data whitening, principal component analysis (PCA) [22] or zero component analysis [27, 20, 46]. However, linear normalization methods might not suffice when the original data distribution is highly non-Gaussian. For tabular and time series data, a popular approach to Gaussianize the marginal distributions is to apply feature-wise non-linear transformations. Two commonly-used parametric methods are the Box-Cox [5] transformation and its extension, the Yeo-Johnson (YJ) transformation [52]. Both have been used in multiple applications, such as climate and weather forecast [53, 50, 51], economics [13] and genomic studies [7, 58, 9].

**Problem and contributions** In this paper, we investigate the problem of data normalization in the cross-silo FL setting, by exploring how to apply the YJ transformation to a distributed dataset. This problem arises frequently in medical cross-silo FL, e.g. when trying to jointly train models on genetic data (see e.g. [19, 57]). Due to data heterogeneity, no single client can act as a reference client: indeed, there is no guarantee that transformation parameters fitted on a single client would be relevant for other clients' data. Hence, it is necessary to fit normalization methods on the full federated dataset. Moreover, in this setting, data privacy is of paramount importance, and therefore FL protocols should be carefully designed. Our main contributions to this problem are as follows:

1. We prove that the negative YJ log-likelihood is convex (Section 3), which is a novel result, to the best of our knowledge.
2. Building on this property, we introduce EXPYJ, a method to fit the YJ transformation based on exponential search (Section 3). We numerically show that this method is more stable than standard approaches for fitting the YJ transformation based on the Brent minimization method [6].
3. We propose SECUREFEDYJ (Section 4), a secure way to extend EXPYJ in the cross-silo FL setting using SMC. We show that SECUREFEDYJ does not leak any information on the datasets apart from what is leaked by the parameters minimizing the YJ negative log-likelihood (Section 4 and Proposition 4.1). By construction, SECUREFEDYJ provides the same results as the pooled-equivalent EXPYJ, regardless of how the data is split across the clients. We check this property in numerical experiments (Section 4). The core ideas behind the resulting algorithm, SECUREFEDYJ, are summarised in Figure 7.

Finally, we illustrate our contributions in numerical applications on synthetic and genomic data in Section 5.

## 2 Background

**The Yeo-Johnson transformation** The YJ transformation [52] was introduced in order to Gaussianize data that can be either positive or negative. It was proposed as a generalization of the Box-Cox transformation [5], that only applies to non-negative data. The YJ transformation consists in applying to each feature a monotonic function $\Psi(\lambda, \cdot)$ parametrized by a scalar $\lambda$, independently of the other features. Thus, there are as many $\lambda$'s as there are features. For a real number $x$, $\Psi(\lambda, x)$ is defined as:

$$\Psi(\lambda, x) = \begin{cases} [(x+1)^\lambda - 1]/\lambda, & \text{if } x \geq 0, \lambda \neq 0, \\ \ln(x+1), & \text{if } x \geq 0, \lambda = 0, \\ -[(-x+1)^{2-\lambda} - 1]/(2-\lambda), & \text{if } x < 0, \lambda \neq 2, \\ -\ln(-x+1), & \text{if } x < 0, \lambda = 2. \end{cases} \tag{1}$$

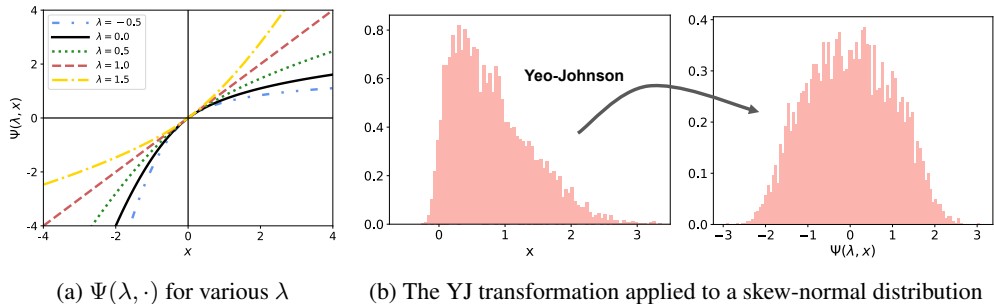

(a) $\Psi(\lambda, \cdot)$ for various $\lambda$  (b) The YJ transformation applied to a skew-normal distribution

Figure 1: The Yeo-Johnson transformation applies a 1-D univariate transform to Gaussianize data.

Figure 1a shows the shape of the YJ function for various values of $\lambda$.

**The Yeo-Johnson likelihood**  Let us consider real-valued samples $\{x_i\}_{i=1,\cdots,n}$, and let us apply the YJ transformation $\Psi(\lambda, \cdot)$ to these samples to Gaussianize their distribution. The log-likelihood that $\{\Psi(\lambda, x_i)\}_{i=1,\cdots,n}$ comes from a Gaussian with mean $\mu$ and variance $\sigma^2$ is given by (derivation details are provided in Appendix A.1):

$$\log \mathcal{L}_{\mathrm{YJ}}(\lambda, \sigma^2, \mu) = -\frac{n}{2}\log(2\pi\sigma^2) - \frac{1}{2\sigma^2}\sum_{i=1}^{n}[\Psi(\lambda, x_i) - \mu]^2 + (\lambda-1)\sum_{i=1}^{n}\mathrm{sgn}(x_i)\log(|x_i|+1).$$

For a given $\lambda$, the log-likelihood is maximized for $\mu_* = \frac{1}{n}\sum_{i=1}^{n}\Psi(\lambda, x_i)$ and $\sigma_*^2 = \frac{1}{n}\sum_{i=1}^{n}(\Psi(x_i, \lambda) - \mu_*)^2$. Once we replace $\mu$ and $\sigma^2$ by $\mu_*$ and $\sigma_*^2$, it becomes:

$$\log \mathcal{L}_{\mathrm{YJ}}(\lambda) = -\frac{n}{2}\log(\sigma^2_{\Psi(\lambda,\{x_i\})}) + (\lambda-1)\sum_{i=1}^{n}\mathrm{sgn}(x_i)\log(|x_i|+1) - \frac{n}{2}\log(2\pi), \quad (2)$$

see [52]. Maximizing the YJ log-likelihood is therefore a 1-dimensional problem for each feature. Once the optimal $\lambda_*$ is found, the transformed data $\Psi(\lambda, x_i)$ is usually renormalized by subtracting its empirical mean $\mu_*$ and dividing by the square root of its empirical variance $\sigma_*^2$. Figure 1b shows an example of the YJ transformation applied to a skew-normal distribution. Note that in a typical application, the triplet $(\lambda_*, \mu_*, \sigma_*^2)$ is fitted on the training data only, and is then used to Gaussianize the test dataset during inference.

**Minimization methods in dimension 1**  As seen above, fitting a YJ transformation can be reduced to a 1D optimization problem. To tackle this problem, we introduce two standard 1D minimization methods: (i) Brent minimization [6] and (ii) exponential search [3].

Brent minimization [6] (not to be confused with the Brent-Dekker method, see [6], chapters 3 and 4) is a widely used method for 1D optimization. It is based on golden section search and successive parabolic interpolations, and does not require evaluating any derivatives. This algorithm is guaranteed to converge to a local minimum with superlinear convergence of order at least 1.3247. Standard implementations of the YJ transformation, in particular the *scikit-learn* implementation [36], are based on the Brent minimization method to minimize the negative log-likelihood provided by Eq. (2).

Exponential search [3] is a dichotomic algorithm designed for unbounded search spaces. The idea is to first find bounds, and then to perform a classic binary search within these bounds. This algorithm can be used to find the minimum of convex differentiable functions with linear convergence, as explained in Appendix B. In this work, we build on exponential search to propose a federated version of the YJ transform, for two main reasons: (i) it is more numerically stable than Brent minimization, as shown in Section 3 and Figure 2, (ii) it may conveniently be adapted to a federated setting, as shown in Section 4, and (iii), this latter federated adaptation offers strong privacy garantees, as shown by Proposition 4.1.

**Secure Multiparty Computation**  As illustrated by various privacy gradient attacks [56, 54], sensitive information on the clients' datasets can be leaked to the central server during an FL training. One way to mitigate this risk is to use Secure Multiparty Computation (SMC) protocols to hide

individual contributions to the server. SMC enables one to evaluate functions with inputs distributed across different users without revealing intermediate results and is often based on secret sharing. SMC protocols tailored for ML use-cases have been recently proposed [12, 14, 34, 39, 48, 33, 49, 41]. These protocols are either designed to enhance the privacy of FL trainings, or to perform secure inference, i.e. to enable the evaluation of model trained privately on a server without revealing the data nor the model.

A popular FL algorithm relying on SMC is Secure Aggregation (SA) [4]. Schematically, in SA each client adds a random mask to their model update before sending it to the central server. These masks have been tailored in such a way that they all together sum to zero. Therefore, the central server cannot see the individual updates of the clients, but it can recover the sum of these updates by adding all the masked quantities sent from them.

More generally, an SMC routine schematically works as follows (we refer to Appendix D for further details). Let us consider the setting where $K$ parties $k = 1, \ldots, K$ want to compute $g = f(h^{(1)}, \ldots, h^{(K)})$ for a known function $f$, where $(h^{(1)}, \ldots, h^{(K)})$ denote private inputs. Each party $k$ knows $h^{(k)}$ and is not willing to share it. During the first step, *secret sharing*, each party splits its private input $h^{(k)}$ into K secret shares $h_1^{(k)}, \ldots, h_K^{(k)}$, and sends the shares $h_{k'}^{(k)}$ to the party $k'$. These secret shares are constructed in such a way that (i) knowing $h_{k'}^{(k)}$ does not provide any information on the value of $h^{(k)}$, and (ii) $h^{(k)}$ can be reconstructed from the vector $(h_1^{(k)}, \ldots, h_K^{(k)})$. For simplicity, we denote $[\![h^{(k)}]\!] = (h_1^{(k)}, \ldots, h_K^{(k)})$ the vector of share secrets. In a second step, *the computation*, each party $k'$ computes the quantity denoted $g_{k'}$ using the secret shares they know along with intermediate quantities exchanged with the other parties. The way to compute $g_{k'}$ depends on $f$ and on the SMC protocol that is used, and is chosen so that $g = f(h^{(1)}, \ldots, h^{(K)})$ can be reconstructed from $(g_1, \ldots g_K)$. Said otherwise, $g_{k'}$ are secret shares of $g$: $[\![g]\!] = (g_1, \ldots g_K)$. Finally, during the *reveal* step, each party $k$ reveals $g_k$ to all other parties, and each party can reconstruct $g$ from $(g_1, \ldots g_K)$.

**Threat model**   In this work, we consider an honest-but-curious setting [35]. Neither the clients nor the server will deviate from the agreed protocol, but each party can potentially try to infer as much information as possible using data they see during the protocol. This setting is relevant for cross-silo FL, where participants are often large institutions whose reputation could be ternished by a more malicious behaviour.

## 3   A novel method to optimize the Yeo-Johnson log-likelihood: EXPYJ

In this section, we leverage the convexity of the negative log-likelihood of the YJ transformation (see Proposition 3.1) to propose a new method to find the optimal $\lambda_*$ using exponential search. While this method only offers linear convergence, compared to the super-linear convergence of Brent minimization method, we demonstrate two of its advantages: (i) it is more numerically stable, and (ii) it is easily amenable to an FL setting with strong privacy guarantees. The method proposed in this section is based on the following result.

**Proposition 3.1.** *The negative log-likelihood* $\lambda \mapsto -\log \mathcal{L}_{\mathrm{YJ}}(\lambda)$ (2) *is strictly convex.*

---

**Algorithm 1** EXPYJ

**Input:** data $x_i$, total data size $n$, number of steps $t_{\max}$
    Initialize $\lambda_{t=0} \leftarrow 0, \lambda_{t=0}^+ \leftarrow \infty, \lambda_{t=0}^- \leftarrow -\infty$
    Compute $S_\varphi$
    **for** $t = 1$ **to** $t_{\max}$
        **for** $g \in \{\Psi(\lambda, \cdot), \Psi(\lambda, \cdot)^2, \partial_\lambda \Psi(\lambda, \cdot), \partial_\lambda \Psi(\lambda, \cdot)^2\}$
            Compute $S_g$
        **end for**
        $\Delta_t = \mathrm{sgn}\left[ nS_{\partial\Psi^2} - 2S_\Psi S_{\partial\Psi} - 2S_\varphi \left(S_{\Psi^2} - \frac{S_\Psi^2}{n}\right)\right]$
        $\lambda_t, \lambda_t^-, \lambda_t^+ \leftarrow \text{EXPUPDATE}(\lambda_{t-1}, \lambda_{t-1}^-, \lambda_{t-1}^+, \Delta_t)$
    **end for**
    $\lambda_* \leftarrow \lambda_{t_{\max}}$
    Compute $\mu_* = S_\Psi/n$ and $\sigma_*^2 = S_{\Psi^2}/n - \mu_*^2$
**Output:** The fitted triplet $(\lambda_*, \mu_*, \sigma_*^2)$

---

The proof of Proposition 3.1 builds upon the work of [26] which shows that the negative log-likelihood of the Box-Cox transformation [5] is convex. The complete proof is deferred to Appendix C.

**The exponential YJ algorithm**   The pseudo-code of the proposed algorithm is presented in Algorithm 1, and relies on the exponential search presented in Algorithm 2 (cf Appendix B for more details on exponential search). An illustration of EXPYJ is shown in Figure 6 in Appendix B. Due to the strict convexity of the negative log-likelihood of the YJ transformation, we may perform the exponential search described in Section 2 and Appendix B. To do so, it is enough to obtain the sign of the derivative. Let $\partial_\lambda \Psi(\lambda, \cdot)^2 = 2\Psi(\lambda, \cdot)\partial_\lambda \Psi(\lambda, \cdot)$ and $\varphi(x) = \text{sgn}(x) + \log(|x| + 1)$. Further, for $g \in \{\Psi(\lambda, \cdot), \partial_\lambda \Psi(\lambda, \cdot), \Psi(\lambda, \cdot)^2, \partial_\lambda \Psi(\lambda, \cdot)^2, \varphi\}$, let us define $S_g \overset{\text{def}}{=} \sum_{i=1}^n g(x_i)$. The derivative of the log-likelihood is available in closed form (see Appendix A.3):

$$\partial_\lambda \log \mathcal{L}_{\text{YJ}} = \frac{n}{2} \frac{S_{\partial \Psi^2} - 2(S_\Psi S_{\partial \Psi})/n}{S_{\Psi^2} - S_\Psi^2/n} - S_\varphi.$$

Notice that $S_{\Psi^2} - S_\Psi^2/n$ can be expressed as a variance, hence is non-negative. We may therefore obtain $\text{sgn}\left[\partial_\lambda \log \mathcal{L}_{\text{YJ}}\right]$ while avoiding performing division by computing

$$\text{sgn}\left[\partial_\lambda \log \mathcal{L}_{\text{YJ}}\right] = \text{sgn}\left[nS_{\partial \Psi^2} - 2S_\Psi S_{\partial \Psi} - 2S_\varphi(S_{\Psi^2} - S_\Psi^2/n)\right]. \tag{3}$$

Avoiding this division is crucial to make the overall procedure more numerical stable, as explained below, and eases the use of SMC routines.

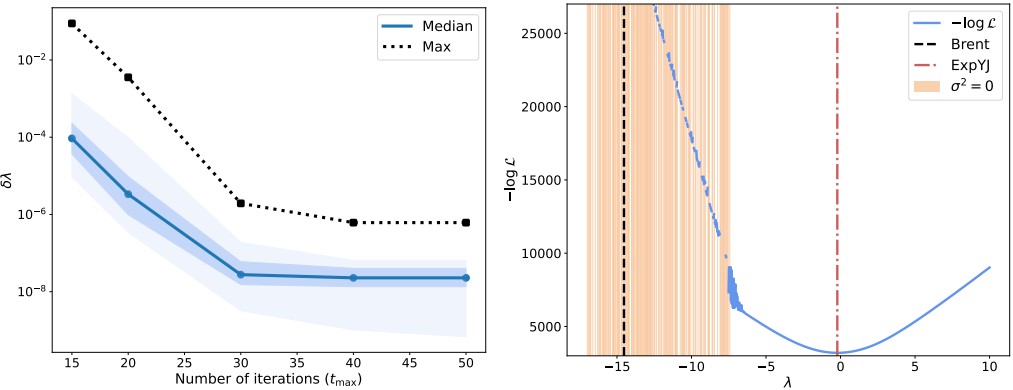

Figure 2: Comparison of EXPYJ and *scikit-learn*. **Left**: For each of the 106 features (see Appendix E.1), we compute the relative difference $\delta\lambda = |\lambda_{\text{EXPYJ}} - \lambda_{\text{sk}}|/|\lambda_{\text{sk}}|$ and plot its median, maximum and 25%-75% and 10%-90% percentiles across the 106 features. **Right**: Negative log-likelihood of the YJ transformation for the mean area of the cell of each sample of the *Breast Cancer* dataset. Full orange bars correspond to values of $\lambda$ for which the likelihood computed using scikit-learn returns $\infty$ as $\sigma_\lambda^2(\{x_i\})$ is equal to 0 up to float-64 machine precision. Dotted lines correspond to the $\lambda_*$ found using Brent minimization or EXPYJ with one client.

**Accuracy of EXPYJ**   We check the accuracy of EXPYJ on the datasets presented in Appendix E.1. In particular, we compare the results provided by EXPYJ with the outputs of the *scikit-learn* algorithm based on Brent minimization.

For 2 of the 108 features present in the datasets, the *scikit-learn* implementation leads to numerical instabilities discussed hereafter. Therefore, we focus our comparison on the 106 remaining features, that we aggregated regardless of the dataset. Figure 2 reports the relative difference $\delta\lambda$ between the results obtained by EXPYJ and by the *scikit-learn* implementation as a function of the number of iteration $t_{\max}$ (as defined in Algorithm 1). These results show that this relative difference is of order less than $10^{-6}$ when $t_{\max} = 40$.

---

**Algorithm 2** EXPUPDATE

---

**Input:** $\lambda, \lambda^+, \lambda^-, \Delta \in \{-1, 1\}$
  **if** $\Delta = 1$ **then**
    $\lambda^- \leftarrow \lambda$
    $\lambda \leftarrow (\lambda^+ + \lambda)/2$ **if** $\lambda^+ < \infty$
    **else** $\lambda \leftarrow \max(2\lambda, 1)$
  **else**
    $\lambda^+ \leftarrow \lambda$
    $\lambda \leftarrow (\lambda^- + \lambda)/2$ **if** $\lambda^- > -\infty$
    **else** $\lambda \leftarrow \min(2\lambda, -1)$
  **end if**
**Output:** Updated $\lambda, \lambda^+, \lambda^-$

---

**Numerical stability of EXPYJ**   Our experiments demonstrate that EXPYJ is numerically more stable than Brent minimization. Indeed, for some values of $\lambda$ and some datasets $\{x_i\}$, the transformation $\Psi(\lambda, \cdot)$ concentrates all data points in a small interval such that the values of $\Psi(\lambda, \{x_i\})$ are all

equal up to machine precision. In that case, the log-likelihood is not well-defined and the term $\log \sigma^2_{\Psi_\lambda}$ takes the value $-\infty$, which prevents Brent minimization from converging. This phenomenon does not affect the EXPYJ routine as we do not compute directly the sign of $\partial_\lambda \mathcal{L} = \partial_\lambda \sigma^2_{\Psi_\lambda} / \sigma^2_{\Psi_\lambda} - \sum_i \varphi(x_i)$, but rather the sign of $\sigma^2_{\Psi_\lambda} \partial_\lambda \mathcal{L} = -\partial_\lambda \sigma^2_{\Psi_\lambda} - \sigma^2_{\Psi_\lambda} \sum_i \varphi(x_i)$, see Eq. (3).

Figure 2 illustrates this in the case of a feature of the *Breast Cancer Dataset*. The $\lambda_*$ returned by the Brent minimization method of *scikit-learn* is $-14.53$ while the minimizer of the negative log-likelihood found by the EXPYJ is $-0.21$. In particular, Figure 2 shows the values of the negative log-likelihood as a function of $\lambda$ computed using 64-bit float precision. The orange vertical full bands correspond to values for which $\sigma^2_{\Psi_\lambda}$ is zero within the machine precision, resulting to a negative log-likelihood of $\infty$. This instability happens for 2 of the 108 features used in numerical experiments, where blindly applying the Brent-based YJ transformation leads to all data points collapsing to zero, while EXPYJ succeeds in transforming the data distributions to more Gaussian-like ones. Appendix E.4 shows that this issue also arises in other real-life datasets.

## 4 Applying the Yeo-Johnson transformation in FL

So far, we only considered the centralized setting, where data is accessible from a single server. Yet, as mentioned in Sections 1 and 2, many real-world situations require working with heterogeneous data split between different centers, and to take privacy constraints into account. When the data is split across centers $k = 1, \ldots, K$ and the function to optimize is separable, i.e. of the form $\mathcal{F}(\lambda) = \sum_{k=1}^{K} f_k(\lambda)$ where each $f_k$ can be computed from data present in the center $k$ exclusively, Federated Learning techniques were recently proposed. In short, they consist in repeatedly performing a few rounds of local optimization in each center, before aggregating local parameters in the server. We refer to [23] for an overview of recent advances in FL. In our case, however, the YJ negative log-likelihood (2) is not separable, due to the log-variance term. Indeed, turning the variance into a separable term would require sharing the global YJ mean $\mu_\lambda$ to all centers at each iteration. Compared to the method we propose in this section, this would lead to more privacy leakage.

We now introduce SECUREFEDYJ, a secure federated algorithm that builds upon EXPYJ to apply YJ transformations. This algorithm satisfies the two following properties: (i) it is *pooled-equivalent*, i.e. it yields the same results as if the data were freely accessible from a single server, and (ii) it leaks as little information as possible about the underlying datasets, as shown by Proposition 4.1. Finally, it converges in a limited number of iterations, thanks to the linear convergence of the underlying exponential search.

**SECUREFEDYJ** SECUREFEDYJ is a federated adaptation of EXPYJ presented in Section 3 to find the best parameters $(\lambda_*, \mu_*, \sigma_*^2)$ of the YJ transformation when training datasets are split across different clients. It relies on SMC to ensure that only the final triplet $(\lambda_*, \mu_*, \sigma_*^2)$ fitted on the training datasets is revealed, without leaking any other information apart from the overall total number of training samples $n$. Indeed, at each intermediate step, only the sign of $\partial_\lambda \log \mathcal{L}_{YJ}$ is revealed, and the mean and variance of the transformed data is only revealed at the last step. The pseudo-code of the resulting algorithm is presented in Algorithm 3, and relies on the exponential search presented in Algorithm 2. A functional representation of SECUREFEDYJ is displayed Figure 7.

In Algorithm 3, we label the clients by $k = 1, \ldots, K$ and each client $k$ holds data $\{x_{k,i} : i = 1, \ldots, n_k\}$. We suppose that the total number of samples $n = \sum_{k=1}^{K} n_k$ is public and shared to all clients. For a given function $g$, we denote $S_{k,g}$ the sum $S_{k,g} \overset{\text{def}}{=} \sum_{i=1}^{n_k} g(x_{k,i})$. As introduced in Section 2, we use double brackets $\llbracket \cdot \rrbracket$ to indicate an SMC secret shared across the clients (see Appendix D for more details).

**Privacy leakage** In Proposition 4.1 we show that Algorithm 3 only reveals information already contained in the fitted triplet $(\lambda_*, \mu_*, \sigma_*^2)$. In comparison, turning the YJ negative log-likelihood (2) into a log-separable function before using off-the-shelf FL methods would require sharing $\mu$ and its gradient and centrally computing $\sigma^2$ for intermediate values of $\lambda$ at each iteration. This could potentially lead to uncontrolled privacy leakage.

**Proposition 4.1.** *The fitted parameter $\lambda_*$ contains all the information revealed during the intermediate steps of* SECUREFEDYJ. *More precisely, there exists a deterministic function $\mathcal{F}$ such*

---

**Algorithm 3** SECUREFEDYJ

---

**Input:** Data $\{x_{k,i}\}$, total data size $n$, number of steps $t_{\max}$
  **Notations:** $[\![\cdot]\!]$ indicates a SMC secret shared across the clients. Any operation such as $[\![\cdot]\!] = f([\![\cdot]\!], [\![\cdot]\!], \cdots)$ where $f$ can be the sum, product, or the sign, designs an SMC routine across the clients as described in Appendix D.5.
  Initialize $\lambda_{t=0} \leftarrow 0, \lambda^+ \leftarrow \infty, \lambda^- \leftarrow -\infty$ independently on each client
  Clients compute in SMC $[\![S_\varphi]\!] = \sum_k [\![S_{k,\varphi}]\!]$
  **for** $t = 1$ **to** $t_{\max}$
    **for** $g \in \{\Psi(\lambda, \cdot), \Psi(\lambda, \cdot)^2, \partial_\lambda \Psi(\lambda, \cdot), \partial_\lambda \Psi(\lambda, \cdot)^2\}$
      Clients compute in SMC $[\![S_g]\!] = \sum_k [\![S_{k,g}]\!]$,
    **end for**
    Clients compute in SMC $[\![\Delta_t]\!] = \mathrm{sgn}\left[n[\![S_{\partial \Psi^2}]\!] - 2[\![S_\Psi]\!][\![S_{\partial\Psi}]\!] - 2[\![S_\varphi]\!]([\![S_{\Psi^2}]\!] - [\![S_\Psi]\!]^2/n)\right]$
    Clients reveal $\Delta_t$
    $\lambda_t, \lambda_t^-, \lambda_t^+ \leftarrow \mathrm{EXPUPDATE}(\lambda_{t-1}, \lambda_{t-1}^-, \lambda_{t-1}^+, \Delta_t)$ independently on each client
  **end for**
  $\lambda_* \leftarrow \lambda_{t_{\max}}$
  Clients compute in SMC $[\![\mu]\!] = \sum_k [\![S_{k,\Psi}]\!]/n$ and $[\![\sigma^2]\!] = \sum_k [\![S_{k,\Psi^2}]\!]/n - [\![\mu^2]\!]$
  Clients reveal $\mu_* \leftarrow \mu$ and $\sigma_*^2 \leftarrow \sigma^2$
**Output:** The fitted triplet $(\lambda_*, \mu_*, \sigma_*^2)$

---

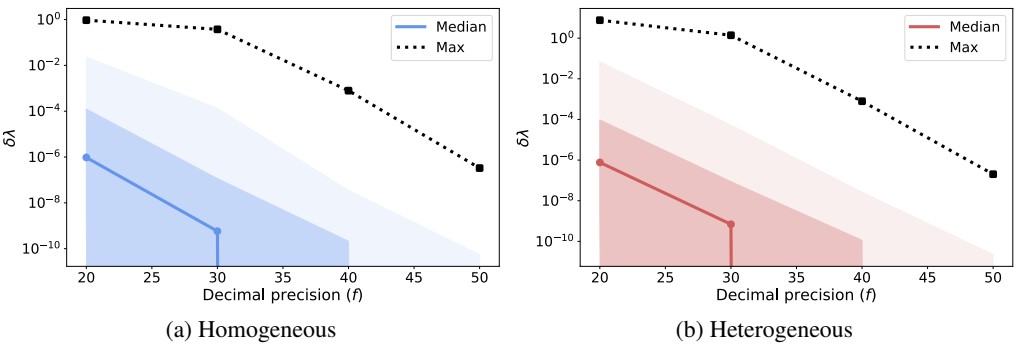

(a) Homogeneous                     (b) Heterogeneous

Figure 3: Comparison of SECUREFEDYJ and EXPYJ for various fixed-point floating precisions $f$ used in SMC, with $l = f + 50$ and $t_{\max} = 40$. The data is distributed across 10 clients, either randomly (homogeneous, *left*), or per decile (heterogeneous, *right*, i.e. each client gets one decile of the data). We report the maximum, median, 25%-75% and 10%-90% percentiles of the relative error $\delta\lambda = |\lambda_{\mathrm{SECFEDYJ}} - \lambda_{\mathrm{EXPYJ}}|/|\lambda_{\mathrm{EXPYJ}}|$ across the 108 features described in Appendix E.1.

*that for any set of datasets $\{x_{k,i}\}$ , if $\lambda_*(\{x_{k,i}\})$ is the result of SECUREFEDYJ on $\{x_{k,i}\}$, then $\{\lambda_t, \lambda_t^+, \lambda_t^-, \Delta_t\}_{t=1,\cdots,t_{\max}} = \mathcal{F}[\lambda_*(\{x_{k,i}\})]$*

**Proof.** This proposition comes from the fact that all gradient signs $\Delta_t$ revealed during the algorithm can be retrospectively inferred from $\lambda_*$. Indeed, $\partial_\lambda \log \mathcal{L}_{\mathrm{YJ}} < 0$ for $\lambda > \lambda_*$ and $\partial_\lambda \log \mathcal{L}_{\mathrm{YJ}} > 0$ for $\lambda < \lambda_*$. Besides, the successive values of $\lambda_t$ explored at each step $t$ can be deterministically inferred from the initial value $\lambda_{t=0}$ and and the final fitted value $\lambda_*$. We construct such a function $\mathcal{F}$ and numerical verify this proposition in Appendix F. ■

**Performance of SECUREFEDYJ**    We implement SECUREFEDYJ in Python, using the MPyC library [42] based on Shamir Secret Sharing [44]. We refer to Appendix D for more details on our implementation. To represent signed real-valued numbers in an SMC protocol, we use a fixed-point representation (see Appendix D.2) using $l$ bits, among which $f$ bits are used for the decimal parts. This means that we consider floats ranging from $-2^{l-f}$ to $2^{l-f}$ and that we have an absolute precision of $2^{-f}$ in our computations.

In order to ensure the accuracy of SECUREFEDYJ results, we need to make sure that $l$ and $f$ are large enough. Figure 3 shows the accuracy of SECUREFEDYJ when compared to EXPYJ for various values of $f$. According to these numerical experiments, taking $f = 50$ and $l = 100$ provides reasonably

accurate results. Moreover, by construction, the outputs of SECUREFEDYJ do not depend on how the data is split across the clients, up to rounding numerical errors. Therefore this algorithm is resilient to data heterogeneity, as long as the numerical decimal precision $f$ is large enough, as shown in Figure 3.

Performing SECUREFEDYJ with $t_{\max} = 40$ takes 726 rounds of communication (see Appendix D.6). During these communication rounds, each client sends overall about 8 Mb per feature to every other client (see also Appendix D.6). SECUREFEDYJ can be applied independently and in parallel to each feature. Therefore, the overall number of rounds does not depend on the number of features being considered, and the communication costs grow proportionally to the number of features. In a realistic cross-silo FL setting as described in [19], the bandwidth of the network is 1 Gb per second with a delay of 20 ms between every two clients. In this context, the execution of SECUREFEDYJ with $t_{\max} = 40$ on $p$ features would take about $726 \times 20$ ms $\simeq 15$ s due to the communication overhead, in addition to $p \times 8$ Mb$/1$ Gbps $\simeq 8p$ ms due to the bandwidth. This shows that SECUREFEDYJ is indeed a viable algorithm in a real-world scenario.

As pointed out in Appendix B, the binary search in the exponential search can be replaced by a $k$-ary search. In such a setting, the sign of the negative log-likelihood of the YJ transformation is computed for $k - 1$ different values of $\lambda$ at each round. Such a modification would reduce the number of communication rounds required to obtain a given accuracy, while increasing the size of the data exchanged over the network at each round.

## 5   Applications

**Genomic data: TCGA**   We start by showing the benefits of YJ preprocessing in survival analysis experiments on lung (LUAD+LUSC), pancreas (PAAD), and colorectal (CRC) cancers. Given gene expression raw counts (features) and censored survival data (responses) from patients having either of those three cancers, we aim to fit a Cox Proportional Hazards (CoxPH) model [10] with the highest possible concordance index (C-index) [25], which measures how well patients are ranked with respect to their survival times. We refer to [25] for a more thorough introduction to survival analysis. In Figure 4, we compare three different preprocessing methods: (i) whitening, (ii) log normalization, and (iii) YJ, each followed by a PCA dimensionality

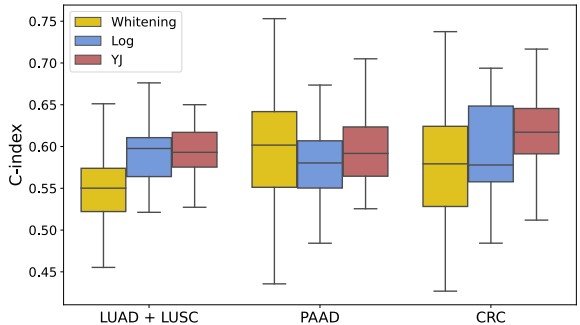

Figure 4: Cross-validation survival analysis performance (higher is better) of a CoxPH model with different normalization methods. The YJ transformation yields either a better or on par performance, and further stabilizes results compared to other approaches.

reduction step. More precisely, whitening (i) consists in centering and reducing to unit variance the total read counts of all genes across all samples, log normalization consists in applying $u \mapsto \log(1 + u)$ to raw read counts before applying global whitening, and YJ is a global YJ transform on the total read counts. We then evaluate each strategy using 5-fold cross-validation with 5 different seeds. We refer to Appendix E.2 for experimental details. While this experiment is performed in a pooled environment, note that, importantly, each step has a federated pooled-equivalent version: apart from the proposed SECUREFEDYJ for YJ, see e.g. [21] for PCA, and Webdisco [31] for Cox model fitting. This simplified setting allows us to understand the importance of the Yeo-Johnson transformation in an ideal setting, independently of other potential downstream federated learning artifacts.

In Figure 4, we see that YJ is better or on par with the best method for each cancer: YJ improves prediction results for colorectal cancer, while yielding results which are on par with the best results for lung and pancreas cancers, with a smaller variance.

**Synthetic data**   We show how applying YJ may help improving performance compared to no or basic preprocessing, and how SECUREFEDYJ yields improvements compared to local YJ transforms

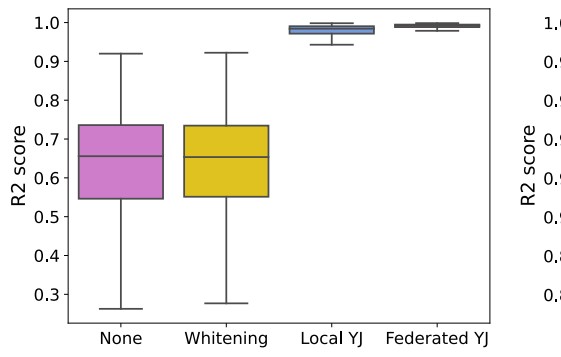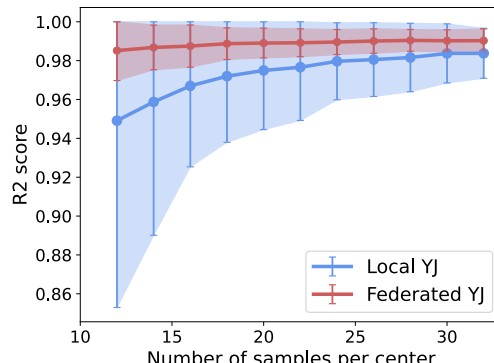

Figure 5: Comparison of different preprocessing methods for linear regression on synthetic federated data. **Left**: performance with 200 samples (20 on each of the 10 centers) on 1000 independent draws, showing the interest of using YJ preprocessing. The R2 score of the models are computed on another dataset of 200 samples not seen during the training. **Right**: Comparison of local and federated YJ over 1000 independent draws. In local YJ, a single center is randomly chosen to fit $\lambda$, which is then shared with other centers, to ensure that the same transformation is applied everywhere. Full lines correspond to the means, error bars to $\pm$ std of the R2 scores of the model on an unseen test dataset.

in federated linear regression. To do so, we generate covariates $\widetilde{X}$ and responses $y$ as

$$
\begin{aligned}
\widetilde{X} &= (\exp(x_1), \exp(x_2 + 2), \sigma(x_3)) \text{ with } X = (x_1, x_2, x_3) \sim \mathcal{N}(0, \mathbf{I}_3), \\
y &= \beta^T X + \varepsilon \text{ with } \varepsilon \sim \mathcal{N}(0, 0.1),
\end{aligned}
\tag{4}
$$

where $\sigma(\cdot)$ is the sigmoid function and $\beta = (-1.3, 2.4, 0.87)$ was randomly chosen. The goal is then to fit a linear model from i.i.d. samples $(\widetilde{X}_i, y_i), i = 1, \ldots, n$ following (4), after an optional preprocessing step. Similarly to the previous example, we simulate a cross-device FL setting only for the preprocessing steps, and the linear model is then fitted in a pooled setting for simplicity. All the details of the numerical experiments are provided in Appendix E.3. We suppose that the samples $(\widetilde{X}_i, y_i), i = 1, \ldots, n$ are homogeneously split across 10 centers. The responses $y_i$ have a highly nonlinear dependency on the covariates $\widetilde{X}_i$, but depend linearly on the $X_i$'s (up to Gaussian noise), which are not observed. Hence, we expect that applying a suitable preprocessing step before training a linear model will transform back the $\widetilde{X}_i$'s into the normally-distributed $X_i$'s and lead to a high performance, compared to no transformation. The results of our experiments are summarized in Figure 5. The left figure shows that the YJ transformation is indeed capable of roughly inverting the $\widetilde{X}_i$'s into the $X_i$'s, yielding a major improvement compared to no preprocessing or standard centering and reduction to unit variance. Besides, the right figure shows that even in this homogeneous setting where the data is i.i.d. across centers, using a federated version of YJ compared to a local version of YJ leads to better average performance, and reduced variance.

## 6 Conclusion

**Summary of our contributions** In this work, we introduce SECUREFEDYJ, a method to fit a YJ transformation on data shared by different clients in a cross-silo setting. SECUREFEDYJ is an SMC version of its pooled equivalent EXPYJ which builds upon the convexity of the negative log-likelihood of the YJ transformation, a novel result introduced by this work, and on the fact that the sign of its derivative can be computed in a stable way. We show that SECUREFEDYJ has the same accuracy as a standard YJ transformation on pooled data. In particular, the results do not depend on how the data is split across the clients, making SECUREFEDYJ resilient to data heterogeneity. Besides, the quantities disclosed by SECUREFEDYJ during the training to the central server do not leak any other information than what is contained in the final parameters $(\mu_*, \lambda_*, \sigma_*^2)$.

**Limitations and future work** While Brent minimization has a super-linear convergence, our approach only has a linear convergence, as it relies on exponential search. This can be an issue if the

communication costs between the clients and the server are high. Acceleration could be achieved by either adapting Brent minimization to a cross-silo setting, or applying a second-order method. We leave the development of a faster SMC methods using either of those two approaches to future work.

Another limitation is that even if our approach reveals only information that would be contained in the final fitted parameters, such parameters themselves might leak information about individual samples, as our approach is not differentially private (DP) [16]. By adding Gaussian or Laplacian noise to each sample's features when computing the $S_g$ terms one could, in principle, make the resulting algorithm DP [1]. However it is unclear to what extent the noise would impact the final accuracy of the method.

Finally, we only consider an *honest-but-curious* setting. We do not explore the threat of a malicious participant that would purposely deviate from the protocol to either gain more information or to jeopardize the convergence. We leave this investigation to future work.

## Acknowledgement

The authors would like to thank the four anonymous reviewers, as well as the anonymous area chair reviewer for their relevant comments and ideas which significantly improved the paper.

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
