# A  Additional properties of the Yeo-Johnson transformation

## A.1  Derivation of the Yeo-Johnson log-likelihood

Using the change of variables rule, the probability to draw a set of points $\{x_i\}$ such that $\{\Psi(\lambda, x_i)\}$ follows a Gaussian distribution of mean $\mu$ and variance $\sigma^2$ is given by:

$$\mathbb{P}(\{x_i\}|\lambda, \mu, \sigma) = \mathbb{P}(\{\Psi_\lambda(\lambda, x_i)\}|\lambda, \mu, \sigma) * \det J[\{x_i\}, \Psi(\lambda, x_i)] \tag{5}$$

where $\det J[x_i, \Psi(\lambda, x_i)]$ is the determinant of the Jacobian matrix $J[\{x_i\}, \{\Psi(\lambda, x_i)\}]$ defined as:

$$J[\{x_i\}, \{\Psi(\lambda, x_i)\}]_{ab} = \frac{\partial \Psi(\lambda, x_a)}{\partial x_b}$$

This matrix is diagonal and each term of its diagonal can be computed using Eq. (1). For each value of $\lambda$ and $x_i$, these diagonal terms can be re-written as $\exp[(\lambda - 1)\operatorname{sgn}(x_i)\log(|x_i| + 1)]$. The term $\mathbb{P}(\{\Psi_\lambda(\lambda, x_i)\}|\lambda, \mu, \sigma)$ is equal to:

$$\mathbb{P}(\{\Psi_\lambda(\lambda, x_i)\}|\lambda, \mu, \sigma) = \prod_i \frac{1}{\sigma\sqrt{2\pi}} \exp\left[-\frac{(x_i - \mu)^2}{2\sigma^2}\right]$$

By taking the logarithm of Eq. (5), we obtain the log-likelihood provided in Section 1 and originally derived on [52].

## A.2  Relationship with the Box-Cox transformation

The Box-Cox transformation [5] works similarly to the YJ transformation, but only applies to strictly positive data. The Box-Cox transformation is based on a function $\Phi(\lambda, \cdot)$ parametrized by $\lambda$ and defined for $x > 0$ as:

$$\Phi(\lambda, x) = \begin{cases} \frac{x^\lambda - 1}{\lambda}, & \text{if } \lambda \neq 0, \\ \ln(x), & \text{if } \lambda = 0. \end{cases}$$

It is straightforward to check that for any $\lambda \in \mathbb{R}$ the YJ transformation $\Psi$ and the Box-Cox transformation $\Phi$ are related by the following equations:

$$\Psi(\lambda, x) = \begin{cases} \Phi(\lambda, x + 1) & \text{if } x \geq 0, \\ -\Phi(2 - \lambda, 1 - x) & \text{if } x < 0. \end{cases}$$

## A.3  Analytical formulae for the derivatives of the Yeo-Johnson transformation

The YJ function is infinitely differentiable with respect to both of its variables ($x$ and $\lambda$). Here are its successive derivatives with respect to $\lambda$:

$$\partial_\lambda^k \Psi(\lambda, x) = \begin{cases} [(x + 1)^\lambda [\ln(x + 1)]^k - k\partial_\lambda^{k-1}\Psi(\lambda, x)]/\lambda, & \text{if } x \geq 0, \lambda \neq 0, \\ \ln(x + 1)^{k+1}/(k + 1), & \text{if } x \geq 0, \lambda = 0, \\ ([-x + 1]^{2-\lambda}[\ln(-x + 1)]^k + k\partial_\lambda^{k-1}\Psi(\lambda, x))/(2 - \lambda), & \text{if } x < 0, \lambda \neq 2, \\ (-\ln(-x + 1))^{k+1}/(k + 1), & \text{if } x < 0, \lambda = 2. \end{cases}$$

# B  Background on exponential search

Exponential search [3] is a method to look for an element in an unbounded sorted array. The idea is to first find bounds on the array such that the element is contained within such bounds, and then perform a classic binary search inside these bounds. Let us consider the task of finding the smallest element $u_{i_0}$ greater than a threshold $C$ in an unbounded sorted array $\{u_i\}_{i \in \mathbb{N}^*}$. The exponential search iteratively looks at $u_i$ for $i \in \{1, 2, 2^2, 2^4, \dots\}$ until it finds a $i_{\max}$ such that $u_{i_{\max}} \geq C$. This takes $\log_2(i_{\max})$ steps. Then it performs a binary search between $i = i_{\max}/2$ and $i = i_{\max}$, which also takes $\log_2(i_{\max}/2)$ steps.

If $f(s)$ is a strictly increasing function of $s$ taking both positive and negative values, one can adapt the exponential search to find the root $s_0$ of $f$. The first step is to find an upper and a lower bound of $s_0$ by evaluating $f$ at different points using an exponential grid (e.g. evaluating $f$ in $s = 1, -1, 2, -2, 2^2, -2^2, 2^4, -2^4, \dots$). Once such bounds are found, one can perform a dichotomic search inside these bounds to find the root $s_0$ of $f$. This dichotomic search has a linear convergence of order 2, with each step summarized in Algorithm 2. It is important to note that this algorithm is correct even if $f$ is not increasing, as long as $f(s) < 0$ when $s < s_0$ and $f(s) > 0$ when $s > s_0$, as is the case in this work when $f$ is the derivative of the negative YJ log-likelihood.

Figure 6 is an illustration of EXPYJ that is based on exponential search.

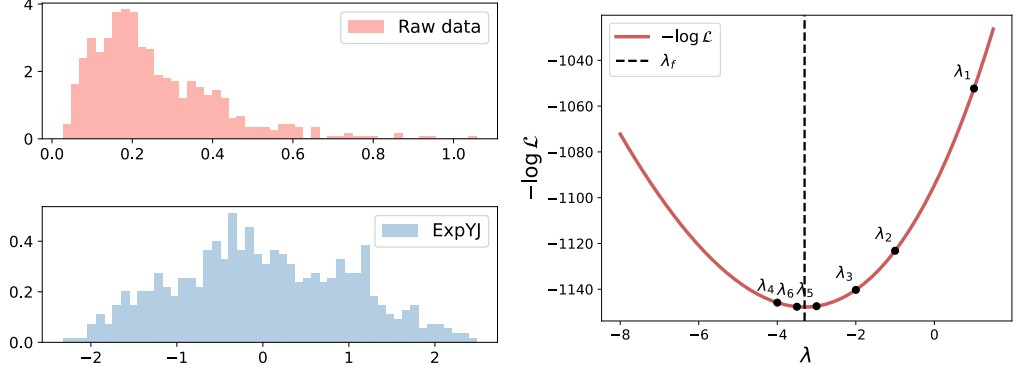

Figure 6: An example of EXPYJ applied to the largest perimeter of the cells in each sample of the *Breast Cancer* dataset. *Left:* histogram of the pooled dataset before (*top*) and after (*bottom*) applying a YJ transformation with fitted parameters. *Right:* negative log-likelihood of the YJ transformation as a function of $\lambda$. The points $\lambda_t$ correspond to the values taken by EXPYJ during exponential search.

A natural extension of the exponential search is to replace the binary search into a $k$-ary search during the dichotomic search. In that case, $k - 1$ values of $f$ are computed at each round. Such a modification reduces the number of steps required for a given accuracy while increasing the number of operations performed at each step.

## C Proof of Proposition 3.1

We first introduce some lemmas that will be required for the main proof.

**Lemma C.1.** *Let $\lambda \mapsto f_i(\lambda)$, $i = 1, \dots, I$ be positive and twice differentiable functions, such that for all $i$, $\lambda \mapsto \ln[f_i(\lambda)]$ is convex. Then $\lambda \mapsto \ln[\sum_i f_i(\lambda)]$ is also convex.*

*Proof.* The proof of Lemma C.1 is based on the following lemma:

**Lemma C.2.** *Let $\{a_i\}_{i=1\dots I}$, $\{b_i\}_{i=1\dots I}$, $(c_i)_{i=1\dots I}$ be real numbers, such that for all $1 \le i \le I$: $a_i \ge 0$, $b_i \ge 0$ and $a_i b_i \ge c_i^2$. Then*

$$\left(\sum_{i=1}^{I} a_i\right)\left(\sum_{i=1}^{I} b_i\right) \ge \left(\sum_{i=1}^{I} c_i\right)^2.$$

Indeed, it holds,

$$\left(\sum_{i=1}^{I} a_i\right)\left(\sum_{i=1}^{I} b_i\right) - \left(\sum_{i=1}^{I} c_i\right)^2 = \left(\sum_{i=1}^{I} a_i b_i - c_i^2\right) + \left(\sum_{i=1}^{I} \sum_{i<j\le I} a_i b_j + a_j b_i - 2c_i c_j\right).$$

The first sum contains only non-negative terms as $\forall i$, $a_i b_i \ge c_i^2$. Recalling that $a_i \ge 0$ and $b_i \ge 0$, the second sum also contains non-negative terms as $a_i b_j + a_j b_i - 2c_i c_j \ge a_i b_j + a_j b_i - 2\sqrt{a_i b_i}\sqrt{a_j b_j} = (\sqrt{a_i b_j} - \sqrt{a_j b_i})^2 \ge 0$

Now let us prove Lemma C.1. The convexity and twice differentiability of $\ln[f_i(\lambda)]$ implies that $\partial_\lambda^2 \ln[f_i(\lambda)] \geq 0$ and therefore that

$$f_i \partial_\lambda^2 f_i - (\partial_\lambda f_i)^2 \geq 0. \tag{6}$$

As $f_i > 0$, we can conclude from Eq. 6 that $\partial_\lambda^2 f_i \geq 0$. Using Lemma C.2 and the linearity of the derivative, we have:

$$\left(\sum_i f_i\right) \partial_\lambda^2 \left(\sum_i f_i\right) - \left(\partial_\lambda \sum_i f_i\right)^2 \geq 0,$$

which means that $\partial_\lambda^2 \ln(\sum_i f_i) \geq 0$.

$\square$

**Lemma C.3.** *Let $\{\alpha_i\}_{i=1,\ldots,n_\alpha}$ and $\{\beta_i\}_{i=1,\ldots n_\beta}$ be two non-empty sets of real numbers, and let us denote $\{\gamma_i\}_i = \{\alpha_1, \ldots, \alpha_{n_\alpha}, \beta_1, \ldots, \beta_{n_\beta}\}$ and $n_\gamma = n_\alpha + n_\beta$. Let $\bar{\alpha} = \frac{1}{n_\alpha}\sum_i \alpha_i$, $\bar{\beta} = \frac{1}{n_\beta}\sum_i \beta_i$, $\bar{\gamma} = \frac{1}{n_\gamma}\sum_i \gamma_i$ and $\sigma_\alpha^2 = \frac{1}{n_\alpha}\sum_i(\alpha_i - \bar{\alpha})^2$, $\sigma_\beta^2 = \frac{1}{n_\beta}\sum_i(\beta_i - \bar{\beta})^2$, $\sigma_\gamma^2 = \frac{1}{n_\gamma}\sum_i(\gamma_i - \bar{\gamma})^2$. Then:*

$$\sigma_\gamma^2 = \frac{n_\alpha}{n_\gamma}\sigma_\alpha^2 + \frac{n_\beta}{n_\gamma}\sigma_\beta^2 + \frac{n_\alpha n_\beta}{n_\gamma^2}(\bar{\alpha} - \bar{\beta})^2.$$

*Proof.* This identity is easily obtained using the definitions of $\sigma_\alpha^2, \sigma_\beta^2$ and $\sigma_\gamma^2$. $\square$

## C.1 Proof of Proposition 3.1

*Proof.* We start by proving the only shows the convexity of $-\log \mathcal{L}_{\mathrm{YJ}}(\lambda)$, and prove strict convexity in Appendix C.4.

Let $\{x_i\}_{i=1\ldots n}$ be our data points and let us split this dataset into non-negative values $\{x_i^+\} = \{x_i | x_i \geq 0\}$ and negative values $\{x_i^-\} = \{x_i | x_i < 0\}$. Let $\gamma_i = \Psi(\lambda, x_i)$, $\alpha_i = \Psi(\lambda, x_i^+)$, and $\beta_i = \Psi(\lambda, x_i^-)$. We denote $n_\alpha, n_\beta, n_\gamma$ the lengths of the sets $\{\alpha_i\}$, $\{\beta_i\}$ and $\{\gamma_i\}$. For clarity, let us consider the case where both $\{x_i^+\}$ and $\{x_i^-\}$ have at least two distinct items and therefore $n_\alpha \geq 1$, $n_\beta \geq 1$ and $\sigma_\alpha^2 > 0$, $\sigma_\beta^2 > 0$. We relegate to Appendix C.3 the other edge cases. According to Lemma C.3, the expression of negative log-likelihood of the YJ transformation provided in Eq. (2) can be reformulated as:

$$-\log \mathcal{L}_{\mathrm{YJ}}(\lambda) = \frac{n}{2}\log(2\pi) - (\lambda - 1)\sum_{i=1}^n \mathrm{sign}(x_i)\log(|x_i| + 1)$$

$$+ \ln\left(\frac{n_\alpha}{n_\gamma}\sigma_\alpha^2 + \frac{n_\beta}{n_\gamma}\sigma_\beta^2 + \frac{n_\alpha n_\beta}{n_\gamma^2}(\bar{\alpha} - \bar{\beta})^2\right). \tag{7}$$

The first term is constant and the second one is linear in $\lambda$ so we only have to prove the convexity of the last term to prove that the full negative log-likelihood is convex. Using Lemma C.1, we only need to show that $\lambda \mapsto \ln\left(\frac{n_\alpha}{n_\gamma}\sigma_\alpha^2\right)$, $\lambda \mapsto \ln\left(\frac{n_\beta}{n_\gamma}\sigma_\beta^2\right)$ and $\lambda \mapsto \ln\left(\frac{n_\alpha n_\beta}{n_\gamma^2}(\bar{\alpha} - \bar{\beta})^2\right)$ are convex. We can get rid of the constant factor and show that $\lambda \mapsto \ln\left(\sigma_\alpha^2\right)$, $\lambda \mapsto \ln\left(\sigma_\beta^2\right)$ and $\lambda \mapsto \ln\left((\bar{\alpha} - \bar{\beta})^2\right)$ are convex.

The key idea of the proof is to use the fact that, according to [26], for any set of positive real numbers $\{a_i\}$, $\lambda \mapsto \ln \sigma[\Phi(\lambda, \{a_i\}]$ is convex, where $\Phi(\lambda, \cdot)$ denotes the Box-Cox transformation. Besides we have (see Appendix A.2):

$$\alpha_i = \Psi(\lambda, x_i^+) = \Phi(\lambda, x_i^+ + 1), \tag{8}$$

$$\beta_i = \Psi(\lambda, x_i^-) = -\Phi(2 - \lambda, 1 - x_i^-). \tag{9}$$

Therefore $\ln \sigma[\alpha_i] = \ln \sigma[\Phi(\lambda, (x_i^+ + 1)]$ which is a convex function of $\lambda$. Similarly, $\sigma[\{-\Phi(2 - \lambda, 1 - x_i^-)\}]^2 = \sigma[\{\Phi(2 - \lambda, 1 - x_i^-)\}]^2$. The function $\lambda \mapsto \sigma[\{\Phi(2 - \lambda, 1 - x_i^-)\}]^2$ is convex as the composition of the linear function $\lambda \mapsto 2 - \lambda$ with the convex function $\lambda \mapsto \sigma[\{\Phi(\lambda, 1 - x_i^-)\}]^2$.

Let us finally prove the convexity of $\lambda \mapsto \ln\left[(\bar{\alpha} - \bar{\beta})^2\right]$. We recall that $\bar{\alpha} > 0$ and $\bar{\beta} < 0$ and that $\ln\left[(\bar{\alpha} - \bar{\beta})^2\right] = 2\ln\left[\bar{\alpha} - \bar{\beta}\right]$. Using Lemma C.1, we only need to prove that $\lambda \mapsto \ln(\bar{\alpha})$ and $\lambda \mapsto \ln(-\bar{\beta})$ are convex. As $\bar{\alpha}$ and $\bar{\beta}$ are defined as sums, still using Lemma C.1, we only need to prove that $\lambda \mapsto \ln\left(\Psi(\lambda, x_i^+)\right)$ and $\lambda \mapsto \ln\left(-\Psi(\lambda, x_i^-)\right)$ are convex for any $i$. Using, Eqs. (8) and (9), it is sufficient to prove that for any real number $a \geq 1$, the function $\lambda \mapsto \ln[\Phi(\lambda, (a)] = \ln[(a^\lambda - 1)/\lambda]$ is convex, which is proved in Appendix C.2.

$\square$

## C.2  Proof that $\lambda \mapsto \ln[\Phi(\lambda, (a)]$ is convex

Let $a \geq 1$ $u(\lambda) = (a^\lambda - 1)/\lambda$ and $g(\lambda) = \ln u(\lambda)$. For $\lambda \neq 0$, the second derivative of $g$ is positive if and only if $D \overset{\text{def}}{=} \lambda^4(uu'' - (u')^2) \geq 0$.

We have

$$D(a, \lambda) = a^{2\lambda} - a^\lambda \lambda^2 \log(a)^2 - 2a^\lambda + 1.$$

Let us show that $D \geq 0$ when $\lambda \neq 0$. $D(a = 1, \lambda) = 0$, so we just need to show that $\partial_a D(a, \lambda) > 0$ when $a > 0$. As

$$\partial_a D(a, \lambda) = a^{(\lambda - 1)}\lambda(2a^\lambda - \lambda^2 \log(a)^2 - 2\lambda \log(a) - 2),$$

let us define $T(a, \lambda)$ as:

$$T(a, \lambda) = (2a^\lambda - \lambda^2 \log(a)^2 - 2\lambda \log(a) - 2).$$

We just need to show that $T(a, \lambda) > 0$ when $\lambda > 0$ and $T(a, \lambda) < 0$ when $\lambda < 0$. As $T(a, 0) = 0$, we just need to show that $\partial_\lambda T(a, \lambda) > 0$ when $\lambda \neq 0$.

$$\partial_\lambda T(a, \lambda) = 2(a^\lambda - \lambda \log(a) - 1)\log(a).$$

As $a > 1$, $\log(a) > 0$, so we just need to show that $(a^\lambda - \lambda \log(a) - 1) > 0$ which can be done by replacing $x$ by $\lambda \log(a)$ in the following inequality: $\exp(x) > x + 1$ for $x > 0$.

To conclude, when $\lambda \neq 0$ and $a \geq 1$, $D(\lambda, a) \geq 0$, and if $a > 1$ then $D(\lambda, a) > 0$. Therefore, the second derivative of $g$ is positive for any $\lambda \geq 0$. Using continuity, we can conclude that the second derivative of $g$ is positive for any $\lambda$ and that $\lambda \mapsto \ln[\Phi(\lambda, (a)]$ is convex.

Note that if $a > 1$, then $D > 0$ and we can conclude that $\lambda \mapsto \ln[\Phi(\lambda, (a)]$ is strictly convex.

## C.3  Edge cases not covered by the main proof of Proposition 3.1

In the main proof we assume that $n_\alpha \geq 2$, $n_\beta \geq 2$ and that $\sigma_\alpha^2 > 0$, $\sigma_\beta^2 > 0$. Said otherwise, we assume that both $\{x_i^+\}$ and $\{x_i^-\}$ have at least two distinct elements. The proof is almost unchanged if this is not the case, as we can discard any term inside the logarithm of Eq. 7. For example, let's assume that $n_\alpha = 1$. Therefore $\sigma_\alpha^2 = 0$. We can then rewrite Eq. 7 as:

$$-\log \mathcal{L}_{\text{YJ}} = \frac{n}{2}\log(2\pi) - (\lambda - 1)\sum_{i=1}^{n} \text{sign}(x_i)\log(|x_i| + 1)$$

$$+ \ln\left(\frac{n_\beta}{n_\gamma}\sigma_\beta^2 + \frac{n_\alpha n_\beta}{n_\gamma^2}(\bar{\alpha} - \bar{\beta})^2\right).$$

We only need to show that $\lambda \mapsto \ln\left(\frac{n_\beta}{n_\gamma}\sigma_\beta^2\right)$ and $\lambda \mapsto \ln\left(\frac{n_\alpha n_\beta}{n_\gamma^2}(\bar{\alpha} - \bar{\beta})^2\right)$ are convex as in the main proof.

Any other edge case can be treated similarly, and the proof holds as soon as $\{x_i\}$ has at least two distinct elements.

## C.4  Strict convexity of the Yeo-Johnson negative log-likelihood.

To prove the strict convexity of the YJ negative log-likelihood, let us notice that under the hypotheses of Lemma C.1, if at least one function $\lambda \mapsto \ln(f_i)$ is strictly convex, then $\lambda \mapsto \ln[\sum_i f_i(\lambda)]$ is

strictly convex. Besides, according to [26], for any set of positive real $\{a_i\}$ with at least two distinct elements, $\lambda \mapsto \ln \sigma[\Phi(\lambda, (a_i)]$ is strictly convex. Therefore, in the case where either $\{x_i^+\}$ or $\{x_i^-\}$ has two distinct elements, we can conclude that the YJ negative log-likelihood is strictly convex.

The only problematic case is when both $\sigma_\alpha^2 = 0$ and $\sigma_\beta^2 = 0$. In that case $\{x_i\}$ has only two distinct element: one positive or null and one strictly negative. In that case, $\lambda \mapsto \sigma[\{\Phi(\lambda, 1 - x_i^-)\}]^2$ is strictly convex as $\lambda \mapsto \ln[\Phi(\lambda, (a)] = \ln[(a^\lambda - 1)/\lambda]$ is strictly convex for $a > 1$.

# D   Secure Multi-Party Computation

## D.1   Shamir Secret Sharing

Secure Multiparty Computation (SMC) consists in evaluating functions without disclosing their inputs. One way to achieve this result is to use secret sharing. The main idea is that a value $h$ is split into different secret shares $h_k$, $k = 1, \cdots, K$ where $K$ is the number of clients. Each client $k$ only knows the value of the secret share $h_k$, and one needs at least $p$ shares with $1 < p \leq K$ to recover the initial value $h$. The set of the secret shares $h_k$ of $h$ is denoted $[\![h]\!]$. Schematically, SMC consists in three main steps: (i) *secret sharing*, where each client splits its input into secret shares and sends them to the other clients (ii) *computation*, where the clients perform mathematical computations on the secret shares and obtain secret shares of the output and (iii) *reveal* steps, where the clients send each other the secret shares of the output in order to reconstruct and reveal the output.

In the Shamir Secret Sharing method [44], the secret shares of $h$ correspond to the values of a given polynomial $P_h(x)$ of order $K$ at different points $x_k$ where $P_h(0) = h$. The values $x_k$ are arbitrarily chosen by the protocol with the constraint that all $x_k$ should be distinct. If all the clients disclose their secret share $h_k = P_h(x_k)$, then the secret $h$ can be recovered by polynomial interpolation. In this framework the addition can be done trivially. If $[\![h]\!] = \{h_k\}_{k=1,\cdots K}$ and $[\![g]\!] = \{g_k\}_{k=1,\cdots K}$ are the shares of $g$, then $[\![g + h]\!] = \{g_k + h_k\}_{k=1,\cdots K}$ are shares of $g + h$. Said otherwise, $[\![g + h]\!] = [\![g]\!] + [\![h]\!]$. Therefore adding two shared secrets requires no communication between the clients. Similarly, multiplying a shared secret by a public constant $c$ is done without communication as $[\![cg]\!] = c[\![g]\!]$. However, multiplying two shared secrets, i.e. computing shares of $[\![gh]\!]$ is more involved and requires one round of communication. More precisely, each client has to send one scalar quantity to all the other clients during this process, as explained for example in [37], section 3.

## D.2   Fixed-Point Representation

The secret shares in SMC belong to a finite set $\mathbb{Z}_p$ where $p$ is a prime number and all the operations are integer operations done modulo $p$. In practice we consider integers encoded using $l$ bits, then we choose the smallest prime number $p$ such that $2^l < p$ and we perform each operation modulo $p$. Therefore any value has to be encoded as an integer using a finite number of bits. To encode negative integers, we consider that encoded integers between 0 and $2^{l-1} - 1$ are positive and encoded integers between $2^{l-1}$ and $2^l - 1$ are negative. We have to choose a value of $l$ large enough such that the highest absolute value considered is below $2^{l-1}$. Real-value numbers are encoded using fixed-point precision, as described in [8], where the $f$ least significant bits of the encoding correspond to the decimal part, and the $l - f$ most significant bits correspond to the integer part. The addition of two fixed-point numbers in SMC can be done as described in Appendix D.1. However, multiplying two fixed-point representation numbers in SMC is more complex as the result must be divided by $2^f$, i.e. the $2^f$ least significant bits are discarded. As explained in detail in [8], multiplying two fixed-point numbers requires two rounds of communication (instead of one round of communication for the multiplication of two integers).

## D.3   Comparison in SMC

In SECUREFEDYJ , we need to compute in SMC the sign of an expression, which is equivalent to making a comparison with 0. As we are using fixed-point representation encoding, computing the sign amounts to computing the most significant bit of the binary decomposition of a given shared secret. In order to do so, we use the method described in [38], which works for any SMC framework supporting addition and multiplication. This method requires 10 rounds of communication among the clients (6 of which can be done offline, i.e. they correspond to random values exchanged

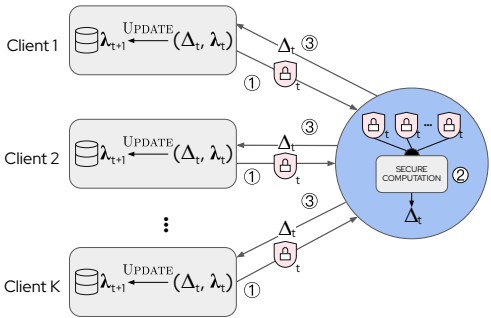

Figure 7: Simplified view of one round of SECUREFEDYJ. ①: All clients compute local, data-dependent quantities. ②: $\Delta_t$ is computed using SMC. Data-dependent quantities computed by each client are not disclosed during the process. ③: $\Delta_t$ is disclosed to all the clients, a new value $\lambda_{t+1}$ is computed using an exponential update.

beforehand and can be done regardless of the value of the input). During these 10 rounds of communication, $153l + 423 \log l + 24$ multiplications are performed, $135l + 423 \log l + 16$ of which can also be done offline. Notice that other SMC primitives could be used, such as the one described in [17] which provides more efficient way to do SMC comparison.

## D.4 MPyC

To implement SECUREFEDYJ we used the python library MPyC [42]. MPyC is based built upon VIFF framework [11] and is based on Shamir Secret Sharing [44]. We refer to [30] for a discussion of the performance of this library for various SMC tasks.

## D.5 Further details on Algorithm 3

The pseudo-code provided in Algorithm 3, is a schematic overview of SECUREFEDYJand relies on the SMC routines described above. For example, the following line of the pseudo-code:

$$[\![S_\varphi]\!] = \sum_k [\![S_{k,\varphi}]\!]$$

implies that: (i) each client $k$ computes $S_{k,\varphi}$, divide it into secrets and send these share secrets to all the other clients; (ii) Using the SMC routines described in Appendices D.1, D.2 and D.4, the clients compute together the share secrets of $[\![S_\varphi]\!]$ where $S_\varphi = \sum_k S_{k,\varphi}$. After this step in Algorithm 3, the value of $S_\varphi$ is therefore shared using share secrets across all the clients. Notice that the server only plays an orchestration roles in this process.

## D.6 Complexity of SECUREFEDYJ

At each step of the exponential search, we share 6 secrets (the values of $S_g$), perform 10 fixed-point multiplications (including multiplying and dividing by $n$), and one comparison (i.e. computing the sign of $\partial_\lambda \mathcal{L}_{\mathrm{YJ}}$.

The 6 secrets can be shared in parallel in one round of communication. Some of the multiplications can also be done in parallel, and only 3 successive rounds of multiplications have to be performed, which require 6 rounds of communications. As stated in Appendix D.3, the comparison requires 10 rounds of communications. Revealing the secret $\Delta$ also requires one round of communication. Notice that the additions do not require any round of communication. This amounts to 18 communications per exponential search step. Besides, computing $[\![S_\phi]\!]$ at the beginning of the algorithm and computing and revealing $\mu_*$ and $\sigma_*^2$ at the end of the algorithm requires 6 more rounds of communication. Overall, performing 40 steps of exponential search with SECUREFEDYJ costs $18 \times 40 + 6 = 726$ rounds of communications.

For each elementary operation, such as sharing a secret, revealing a secret or making a multiplication, the order of magnitude of the size of the message sent by each client to the other clients is $\lceil \log_2(p) \rceil$

bits. Notice that $\log_2(p)$ is of the same order of magnitude of $l$ as $p$ is the smallest prime number above $2^l$. More precisely, each client sends around $l$ bits to each of the other clients for these elementary operations. The overall size of the messages exchanged during the 726 rounds of communications mentioned above is mainly dominated by the $153l + 423 \log l + 24$ multiplications done at each of the 40 comparisons. Taking $l = 100$, we find that each client sends overall around $6.5 \, 10^7$ bits (or $\sim 8$ Mega-bytes) to each of the other clients during SECUREFEDYJ.

# E    Details of the numerical experiments

## E.1    Datasets used in this work

**Datasets exposed by** *scikit-learn* **API used in Figure 2 and Figure 3**    For numerical experiments, we use four public datasets available in the UC Irvine Machine Learning repository [15] under a Creative Commons Attribution 4.0 International (CC BY 4.0) license and exposed by the *scikit-learn* datasets API. These datasets are the *Iris dataset* [18] (150 samples, 4 features), the *Wine Data Set* (178 samples, 13 features), the *Optical Recognition of Handwritten Digits Data Set* (1797 samples, 64 features) and the *Breast Cancer Wisconsin (Diagnostic) Data Set* (569 samples, 30 features). Only keeping features that have at least two distinct values, these datasets provide a total of 108 different features.

**Extra UC Irvine Machine Learning repositories used to test Brent minimization method**    We used 19 extra datasets in Appendix E.4, in order to test the instabilities of the Brent minization method applied to the YJ transformation. All these datasets are present in [15] under a Creative Commons Attribution 4.0 International (CC BY 4.0).

**Genomic data used in Figure 4**    For genetic experiments, we rely on RNA-seq expression data from The Cancer Genome Atlas, expressed in Fragments per Kilobase Million (FPKM). We focus on 3 cancers: colorectal cancer (COAD), lung cancer (LUAD + LUSC), and pancreatic adenocarcinoma (PAAD). These datasets are available on `https://portal.gdc.cancer.gov/` under Open Access.

## E.2    Experiments on TCGA data

Based on FPKM counts, we load all available data for each cancer of interest, removing genes with null expression for all samples.

**Pipeline**    Our pipeline consists of three steps:

1. Normalization: either *whitening*, *log*, or Yeo-Johnson transformation;

2. Dimensionality reduction: a PCA was applied on normalized data to reduce dimension (dimension 128 for lung and colorectal cancer, 90 for pancreatic cancer);

3. Cox Proportional Hazards (CoxPH) [10] model fitting.

**Normalization**    All normalization steps are performed on counts, regardless of the genes, as counts are related to the same underlying phenomenon induced by next-generation RNA sequencing. In other words, for the plain whitening, a single mean and variance is computed. For *log*, following application of $\log(1 + \cdot)$ to all entries, a similar count-level whitening is performed. For the YJ transformation, we perform 10 iterations of the proposed algorithm.

**CoxPH model training**    CoxPH models are fitted with *lifelines* (0.26.4). We use an $\ell_2$ regularization of magnitude 10 for each cancer, without any hyperparameter optimization.

**Cross-validation**    Results are computed following 5-fold stratified group cross-validation, repeated 5 times with different seeds. Stratification is performed to ensure a balanced set of censored patients in each fold, while ensuring that samples belonging to the same patients end up in the same group to avoid over-estimating the generalization of the model.

| Dataset name | # of samples | # of features (with at least two distinct values) | # of instabilities of Brent minimization |
|---|---|---|---|
| airfoil self noise | 1503 | 5 | 0 |
| blood transfusion | 748 | 4 | 1 |
| breast cancer diagnostic | 569 | 30 | 2 |
| climate model crashes | 540 | 18 | 0 |
| concrete slump | 103 | 7 | 0 |
| connectionist bench sonar | 208 | 60 | 0 |
| connectionist bench vowel | 990 | 10 | 0 |
| ecoli | 336 | 7 | 2 |
| glass | 214 | 9 | 0 |
| ionosphere | 351 | 34 | 0 |
| iris | 150 | 4 | 0 |
| libras | 360 | 90 | 0 |
| parkinsons | 195 | 23 | 0 |
| planning relax | 182 | 12 | 0 |
| qsar biodegradation | 1055 | 41 | 0 |
| seeds | 210 | 7 | 0 |
| wine | 178 | 13 | 0 |
| wine quality red | 1599 | 10 | 0 |
| wine quality white | 4898 | 11 | 0 |
| yacht hydrodynamics | 308 | 6 | 0 |
| yeast | 1484 | 8 | 0 |

Table 1: Number of feature for which the *scikit-learn* implementation of Yeo-Johnson based on Brent minimization method fails for 21 different datasets available on the UC Irvine Machine Learning repository [15]. We only kept the features with at least two distinct values.

### E.3 Experiment on synthetic data

To generate the results of Section 5, we sampled for each of the 10 centers 200 datapoints using Eq. (4). We then apply an optional preprocessing steps before fitting a linear regression model using scikit-learn *LinearRegression* model on the pooled data. Another dataset of 200 points was then generated, and we computed the R2 on this unseen dataset. This experiment was repeated 1000 times using each time a different seed and the box plot in Section 5 presents the min-max, the median the first and the third quartile. The different preprocessing steps shown are:

- None: no preprocessing step is applied
- Whitening: for each of the three dimensions of $X_i$, we subtract the empirical mean and we divide by the empirical standard deviation computed across all ten centers to the train dataset and the test dataset
- LocalYJ: we use one center randomly chosen to perform EXPYJ with $t_{\max} = 20$ to each of the dimensions of the dataset. The fitted triplets $\lambda_*, \mu_*, \sigma_*^2$ found for each column are then used to normalize the dataset of all 10 centers and the test dataset.
- Federated YJ: We apply SECUREFEDYJ with $t_{\max} = 20$ on the 10 centers to each of the dimensions of the dataset. The fitted triplets $\lambda_*, \mu_*, \sigma_*^2$ found for each column are then used to normalize the dataset of all 10 centers and the test dataset.

### E.4 Testing Brent minimization on more dataset

As explained in the paragraph *Numerical stability of* EXPYJ of Section 3, applying blindly the Brent minimization method of scikit-learn to minimize the Yeo-Johnson negative log-likelihood might result in numerical instabilities and might collapse all the values of the dataset into a single value. To check further whether this phenomenon is likely to appear, we apply the scikit-learn Yeo-Johnson transformation to various real-life tabular datasets that are either from the UC Irvine Machine Learning repository [15] (which are under a Creative Commons Attribution 4.0 International, CC BY 4.0). For each dataset, we only kept the features that have at least two distinct values. We found that for the 409 features out of 21 datasets, this issue arises 5 times, as summarized by Table 1

## F  Further details on Proposition 4.1

Proposition 4.1 states that all intermediate quantities of SECUREFEDYJ can be recovered from its final result $\lambda_*$. We provide in Algorithm 4 a way to construct the function $\mathcal{F}$ introduced in Proposition 4.1 that can perform this recovery.

We apply Algorithm 4 on the 108 features used in Figure 3, with a fixed-point precision of $f = 50$. We numerically check that the output of $\mathcal{F}$ from Algorithm 4 matches the intermediate quantities revealed by Algorithm 3 up to machine precision.

**Algorithm 4** Function $\mathcal{F}$ recovering quantities revealed by SECUREFEDYJ

**Input:** Hyperparameters $\lambda_{t=0}, \lambda_{t=0}^-, \lambda_{t=0}^+$ number of steps $t_{\max}, \lambda_*$
  **for** $t = 1$ **to** $t_{\max}$
    **if** $\lambda_{t-1} < \lambda_*$ **then**
      $\Delta_t = 1$
    **else**
      $\Delta_t = -1$
    **end if**
    $\lambda_t, \lambda_t^-, \lambda_t^+ \leftarrow \text{EXPUPDATE}(\lambda_{t-1}, \lambda_{t-1}^-, \lambda_{t-1}^+, \Delta_t)$
  **end for**
**Output:** $(\lambda_t, \lambda_t^-, \lambda_t^+, \Delta_t)_{t=0,\ldots,t_{\max}}$