# OpenReview forum: "SecureFedYJ: a safe feature Gaussianization protocol for Federated Learning"
_NeurIPS.cc/2022/Conference — NeurIPS 2022 Accept_

### Official Review · Reviewer_5G9f · 2022-06-30

**Rating:** 6
**Confidence:** 4
**Soundness:** 4 excellent
**Presentation:** 3 good
**Contribution:** 3 good

**Summary:**

The paper addresses the Yeo-Johnson feature transformation, and how it can be computed in the federated setting, where privacy and security restrict how information can be shared. The paper demonstrates the convexity of the log-likelihood function, which enables an efficient search procedure in the multiparty computation model that does not leak any information beyond the revealed answer.  Experiments are included that show the behavior inpractice.

**Questions:**

An alternative approach in the cross silo setting would before each client to apply the YJ transformation locally to their data, on the assumption that the distribution is common across clients.  Can you show examples that demonstrate where this heuristic will fail?



**Limitations:**

The conclusions section has a nice discussion of limitations and future work that is appropriate for the paper.

**Strengths And Weaknesses:**

Strengths

- There is growing interest in secure machine learning that goes beyond the core training part of the ML pipeline, and provides security for data preparation in addition

- The paper shows a nice result about the convexity of the Yeo-Johnson transformation which enables its secure computation

- The proof of minimum leakage (proposition 4.1) is a very nice contribution, and clearly makes the argument that there is no overall leakage from the algorithm

Weaknesses

- The paper does not offer insights that go beyond the immediate question of the YJ transformation.

- The search algorithm requires some operations which are potentially costly to perform in the SMC setting

- The paper glosses over the technical details of applying the algorithm in the SMC setting: there are some moderately complex polynomial operations on data that require multiplications / exponentiations, but the cost of these is not discussed in much detail -- Appendix D gives only a very high level overview of basic secret sharing.  Moreover, the paper does not indicate where the challenge lies: once algorithm 1 is presented, it seems that Algorithm 3 is very similar except that data access is performed through secret sharing.

- The communication overhead is high (8MB per client per feature), and there is no discussion of whether this could be reduced or traded off  -- this is why the method is motivated in the cross-silo rather than cross-device FL setting

---

> ### Author Response · Authors · 2022-08-02
> **Answer to reviewer 5G9f**
>
> We would like to thank anonymous reviewer 5G9f for their review. We answer below all their remarks and questions.
>
> ## Remark 1
>
> The paper does not offer insights that go beyond the immediate question of the YJ transformation.
>
> ## Answer to remark 1
> We are not sure what the reviewer meant by this remark. The goal of this paper is indeed to propose a secure adaptation of the YJ transformation, which is widely applied in e.g., applications to genomics, to the FL setting.
>
> ## Remark 2
> The search algorithm requires some operations which are potentially costly to perform in the SMC setting.
>
> ## Answer to remark 2
> First of all, our numerical experiments based on MPyC reproduces all the SMC computations on a single computer. We therefore measure computation costs, and only the cost of communicating over the network is not simulated.
> Besides, we provide in appendix D.4 the communication cost and the computational complexity of the different SMC operations performed in our algorithm using the SMC routines from [36] and implemented in MPyC. Using these numbers, we compute at the end of section 4, in the paragraph starting line 261 in the revised version (line 264 in the first version), the cost of the whole routine on real cross-silo FL project. We conclude therefore in the paper that:
>
> > In a realistic cross-silo FL setting [...] the execution of secFedYJ with $t_\mathrm{\max} = 40$ on $p$ features would take about $726\times 20\ \mathrm{ms} \simeq 15\ \mathrm{s}$ due to the communication overhead, in addition to $p\times8\ \mathrm{Mb} / 1\ \mathrm{Gbps} \simeq 8 p\ \mathrm{ms}$ due to the bandwidth.
>
>
> ## Remark 3
> The paper glosses over the technical details of applying the algorithm in the SMC setting: there are some moderately complex polynomial operations on data that require multiplications / exponentiations, but the cost of these is not discussed in much detail -- Appendix D gives only a very high level overview of basic secret sharing. Moreover, the paper does not indicate where the challenge lies: once algorithm 1 is presented, it seems that Algorithm 3 is very similar except that data access is performed through secret sharing.
>
> ## Answer to remark 3
> - Regarding the cost of the SMC routines, we refer to our answer of the previous remark.
>
>
> - We thank the reviewer for pointing out that the challenges and the details of Algo 3 were not detailed enough. We modified the pseudo-code and added the appendix D.5 to provide more details about the role of each party and how SMC is used in Algo3. In this algorithm, the server only plays the role of the orchestrator. All the quantities in bracket are shared-secret quantities computed across the client using SMC routines. The new version of the article explicitly details the roles of each party.
>
> ## Remark 4
> The communication overhead is high (8MB per client per feature), and there is no discussion of whether this could be reduced or traded off -- this is why the method is motivated in the cross-silo rather than cross-device FL setting.
>
> ## Answer to remark 4
> We consider the cross-silo FL where a good bandwidth exists between actors. secFedYJ is a solution that can provide a pooled-equivalent result with reasonable communication cost. In this setting, trying to reduce the communication costs might incur a drop in performance, which is not necessary given the setup.
> We agree that in other settings, such as cross-device FL, adjusting the trade-offs would be very relevant. However, this is not the scope of the paper, and we leave such a question for future work.
>
> ## Remark 5
> An alternative approach in the cross silo setting would before each client to apply the YJ transformation locally to their data, on the assumption that the distribution is common across clients. Can you show examples that demonstrate where this heuristic will fail?
>
> ## Answer to remark 5
> Indeed, under the restrictive assumption that the distribution is common across all clients, applying local YJ transformations will yield the expected result. However, this heuristic would fail if clients have *heterogeneous* data distributions. secFedYJ can be applied in both cases, without any check on the data distribution as its result does not depend on the way the data is distributed across the clients.

---

> > ### Comment · Reviewer_5G9f · 2022-08-04
> > **Thanks**
> >
> > I thank the authors for addressing my questions, and clarifying some aspects of their work.
> >
> > > The paper does not offer insights that go beyond the immediate question of the YJ transformation.
> >
> > This is a factual description of the content of the paper.  The results apply to the YJ transformation, but the paper does not suggest any other transformations or functions that this approach could be used for.

---

### Official Review · Reviewer_Kq6t · 2022-07-01

**Rating:** 5
**Confidence:** 3
**Soundness:** 3 good
**Presentation:** 2 fair
**Contribution:** 2 fair

**Summary:**

The authors propose ExpYJ, a new algorithm for feature Gaussianization which beats Brent algorithm in terms of numerical stability but asymptotically slower than Brent algorithm. The authors them combine ExpYJ with secure aggregation to obtain FedYJ.

**Questions:**

1. Please clarify how the algorithm is customized to deal with heterogeneity if any?
2. Have the authors considered discretization in SMC when performing the empirical evaluation?



**Limitations:**

1. Please include empirical evaluation of computation and communication cost on real clusters.
2. Please add details about discretization for SMC.
3. Please validate the numerical stability advantage on more real-world datasets.
4. Please refine the proof for Proposition 4.1 (for example, simulation-based proof).
5. Please clarify how the algorithm is customized to deal with heterogeneity if any?

**Strengths And Weaknesses:**

The authors propose a federated feature Gaussinaization algorithm by first proposing a new optimization algorithm to optimize the hyper-parameters for YJ transformation and then combining it with secure aggregation in FL.

Strength:
1. The paper is well written and easy to follow.
2. The authors evaluate their algorithm on several downstream applications.
3. The improvement on numerical stability is a good contribution. However, I think evaluation on only one dataset is not enough. I encourage the authors to validate this advantage on more datasets.

Weakness:
1. The technical novelty is limited. The proof of strict convexity is a contribution, but with the convexity of Box-Cox transformation, the proof is not too challenging.
2. FedYJ requires multiplication on secret shared values so benchmarking the system performance (how long does it take to perform the computation and communication) is a necessity.
3. The authors highlight the heterogeneity challenge in the introduction but I don't see how they customize their algorithm to deal with heterogeneity. If I understand correctly, the application itself is not sensitive to heterogeneity. For example, I cannot tell any difference between Figure 3(a) and 3(b).
4. The threat model is too weak, which makes the whole setting degenerate to distributed learning instead of FL.
5. Proposition 4.1 is correct but the proof is too sketchy.
6. To use SMC, the data points need proper discretization. The authors do not mention any details about the discretization process like the hyper-parameters.
7. Some minor points: 7.1 There are many references to Appendix C. I recommend the authors to refer to subsections in Appendix C which will make the flow clearer. 7.2 The secure multiparty computation background is almost taking up a whole page. Consider moving it to the appendix.

---

> ### Author Response · Authors · 2022-08-02
> **Answer to Reviewer Kq6t, part 1/2**
>
> We would like to thank anonymous reviewer Kq6t for their review. We answer below all their remarks and questions, and we modify the paper accordingly. We think that thanks to the reviewer remarks, the paper is now clearer, and we would like to politely ask the reviewer to re-evaluate the revised paper. We are open to further discussion with the reviewer regarding the revised paper.
>
> ## Remark 1
>
> FedYJ requires multiplication on secret shared values so benchmarking the system performance (how long does it take to perform the computation and communication) is a necessity. (Also linked with: Please include empirical evaluation of computation and communication cost on real clusters.)
>
> ## Answer to remark 1
> Figure 3 displays the results of a numerical benchmark of the SMC routines on a single computer, thereby measuring computation time. Only the cost due to network communication was not taken into account. However, in order to provide a reasonable communication benchmark, we provide in appendix D4 both the algorithmic complexity of the full SMC routine, and the number of messages sent on the network when performing secFedYJ. Using the bandwidth values of a real-life cross-silo FL project, we estimated at the end of section 4 line 261 of the revised version (line 264 of the original version), that:
> > In this context, the execution of secFedYJ with $t_\mathrm{\max} = 40$ on $p$ features would take about $726\times 20\ \mathrm{ms} \simeq 15\ \mathrm{s}$ due to the communication overhead, in addition to $p\times8\ \mathrm{Mb} / 1\ \mathrm{Gbps} \simeq 8 p\ \mathrm{ms}$ due to the bandwidth. This shows that secFedYJ is indeed a viable algorithm in a real-world scenario.
> We leave a large-scale evaluation of our method on a real cluster to future work.
>
>
>
> ## Remark 2
> The threat model is too weak, which makes the whole setting degenerate to distributed learning instead of FL.
> ## Answer to remark 2
> In this work, we consider, cross-silo FL with a honest-but-curious threat model. Such a threat model is relevant to consider in FL for the following reasons:
>
> - First of all, projects including few organizations that collaborate to train a model together, is indeed in the scope of cross-silo  Federated Learning. Indeed, as stated by “Advances and Open Problems in Federated Learning” [1]: Since the term federated learning was initially introduced with an emphasis on mobile and edge device applications interest in applying FL to other applications has greatly increased, including some which might involve only a small number of relatively reliable clients, for example multiple organizations collaborating to train a model. We term these two federated learning settings “cross-device” and “cross-silo” respectively.
>
> - In cross-silo FL, participants are often large institutions or companies dealing with regulated data, which undergo frequent audits and desire to maintain reputation. An example of such a setting is the MELLODDY project [4], where each partner runs a code that has been previously audited. The honest-but-curious model is therefore a relevant threat model here.
> - Besides, such a threat model have been often considered and studied in several related cases:
> In [3], the author stresses the relevance of this model (also called semi-honest): “in many settings, one may assume that although the users may wish to cheat, they actually behave in a semi-honest way” and underlines interest of this semi-honest model from both a theoretical and practical standpoint, given the difficulty to hack code: “In addition to the methodological role of semi-honest parties in our exposition, they do constitute a model of independent interest. In particular, deviation from the specified program, which may be invoked inside a complex software application, is more difficult than merely recording the contents of some communication registers”.
> - In [2], the authors take the example of a Smart grid project where the energy provider has access to the private energy consumption of users. This work states that: “However, a legitimate participant in the protocol such as the energy supplier could not realistically be modeled as a DY [ = malicious] adversary. In reality, various factors limit the capabilities of the energy supplier including regulations, audits, oversight and desire to maintain reputation.” [...] “We therefore propose to model this agent as a semi-honest or honest-but-curious”
>
>
> [1] Kairouz, P., McMahan, H. B., Avent, B., Bellet, A., Bennis, M., Bhagoji, A. N., ... & Zhao, S. (2021). Advances and open problems in federated learning. Foundations and Trends® in Machine Learning, 14(1–2), 1-210.
>
> [2]Andrew Paverd, Andrew Martin, and Ian Brown. Modelling and automatically analysing privacy properties for honest-but-curious adversaries. Tech. Rep, 2014.
>
> [3]  Foundations of cryptography, Oded Goldreich, Volume 2, page 619, sec 7.2.2
>
> [4] MELLODDY project, www.melloddy.eu

---

> > ### Author Response · Authors · 2022-08-02
> > **Answer to Reviewer Kq6t, part 2/2**
> >
> > ## Remark 3:
> > The authors highlight the heterogeneity challenge in the introduction but I don't see how they customize their algorithm to deal with heterogeneity. If I understand correctly, the application itself is not sensitive to heterogeneity. For example, I cannot tell any difference between Figure 3(a) and 3(b). Also linked with: Please clarify how the algorithm is customized to deal with heterogeneity if any?
> >
> > ## Answer to remark 3:
> > We claim that the outcome of secFedYJ does not depend on how the data is distributed across the clients, and that its result would be the same if all the data were pooled in a central server. Therefore, we do not have to customize secFedYJ when facing an heterogeneous data distribution across clients. This point is illustrated in Fig 3(a) and 3(b), in which the same algorithm is used on data that is either homogeneously or heterogeneously distributed across the center. In particular, note that the result of the algorithm is the same in both cases.
> >
> >
> > ## Remark 4
> > Proposition 4.1 is correct but the proof is too sketchy.
> > Please refine the proof for Proposition 4.1 (for example, simulation-based proof).
> >
> > ## Answer to remark 4
> > We thank the reviewer for this excellent remark. In order to refine this proof, we added in Appendix F an explicit pseudo-code to build such a function F. We numerically checked that this function F recovers the intermediate quantities from $\lambda_*$ as stated by the proof.
> > To use SMC, the data points need proper discretization. The authors do not mention any details about the discretization process like the hyper-parameters.
> >
> > ## Remark 5
> > Please add details about discretization for SMC.
> > Have the authors considered discretization in SMC when performing the empirical evaluation?
> >
> > ## Answer to remark 5
> > Indeed, as pointed out by the reviewer, all quantities are discretized when using SMC. All the floating quantities are discretized in our work using fixed-point representation as described in appendix D.2. The discretization is parametrized by $l$ and $f$ which are respectively the total number of bits to encode each quantity, and the number of bits dedicated to the decimal part of float number. We show the impact of this discretization on the performance of our algorithm in figure 3 where the x-axis corresponds to the discretization parameter $f$.
> >
> > ## Remark 6
> > Please validate the numerical stability advantage on more real-world datasets.
> >
> > ## Answer to remark 6
> > We added in the appendix E.4  a table checking whether this stability appears on more datasets. Out of 484 features, we identified 5 instabilities.
> >
> > ## Remark 7
> > Some minor points: 7.1 There are many references to Appendix C. I recommend the authors to refer to subsections in Appendix C which will make the flow clearer. 7.2 The secure multiparty computation background is almost taking up a whole page. Consider moving it to the appendix.
> >
> > ## Answer to remark 7
> > We thank the reviewer for this suggestion.  There also was a typo in appendix C where two paragraphs were inverted. We fixed it, and we clarified the reference to appendix C.

---

> > > ### Comment · Reviewer_Kq6t · 2022-08-06
> > > **Thanks**
> > >
> > > I would like to thank the authors to answer my questions and address some of my concerns. I would increase my score accordingly and take the response into consideration when discussing with the other reviewers and the AC.

---

> > > > ### Author Response · Authors · 2022-08-08
> > > > **Answer to reviewer Kq6t**
> > > >
> > > > Dear reviewer Kq6t,
> > > >
> > > > We thank you for considering our answers and for increasing the score of your review.

---

### Official Review · Reviewer_W31E · 2022-07-06

**Rating:** 4
**Confidence:** 4
**Soundness:** 3 good
**Presentation:** 3 good
**Contribution:** 3 good

**Summary:**

This paper proposes an exponential search method for YJ transformation in centralized and federated learning settings. It proved that the negative log-likelihood of YJ transformation is strictly convex with respect to \lambda. Building upon that, they provide a method to do an exponential search to find the best \lambda, \mu, and \sigma for YJ transformation. The paper also applies the Secured Multiparty Computation to provide a privacy-preserving guarantee in the process of Federated YJ transformation.

**Questions:**

- In Algorithm 3, who will run the algorithm, and what is the job of the clients and the server?
- What is the reason you only consider cross-silo FL since, in my opinion, this method can be generalized to all kinds of FL.
- Can you please explain the log-likelihood in Eq. 1? I followed reference 50 in the text, but there's no information in the reference.
- What is the advantage of the proposed method for FL?

**Limitations:**

- Unclear threat model
- The proposed method will destroy the heterogeneity of Federated Learning.
- There's no theoretical guarantee of the clients' data privacy leakage from the output of the proposed method.

**Strengths And Weaknesses:**

Strengths:
- Provide a novel exponential search method for YJ transformation.
- The proposed method achieves high accuracy compared to Brent minimization, which is widely used.
- The proposed method has better numerical stability than existing methods (Brent minimization).

Weakness:
- Unclear threat model: can not understand the purposes or capabilities of the threat model.
- The proposed method will destroy the heterogeneity of Federated Learning: After the YJ transformation process, it seems like every client will have the same distribution since they share the same \lambda_*, \mu_*, \sigma_*. This poses a risk to FL: if the adversary knows the data distributions of every client (since they have the same distribution), the adversary will quickly deploy the attack of the tails [1]. Moreover, since the threat model is "honest-but-curious", using the YJ transformation will pose a privacy risk to the clients' data since the server can deploy the membership inference attacks [2].
- There's no theoretical guarantee of the clients' data privacy leakage from the output of the proposed method: since "the fitted parameter \lambda_* contains all the information revealed during the intermediate steps," whether it can leak any information and whether we should protect it?

[1] Wang, Hongyi, et al. "Attack of the tails: Yes, you really can backdoor federated learning." Advances in Neural Information Processing Systems 33 (2020): 16070-16084.

[2] Shokri, Reza, et al. "Membership inference attacks against machine learning models." 2017 IEEE symposium on security and privacy (SP). IEEE, 2017.

---

> ### Author Response · Authors · 2022-08-02
> **Answer to reviewer W31E Part 1/2**
>
> We would like to thank anonymous reviewer W31E for their time and their thorough review. We answer below all their remarks and their questions, and we modified the paper accordingly. We think that thanks to the reviewer's remarks, the paper is now clearer, and we would like to politely ask the reviewer to re-evaluate the revised paper. We are open to further discussion with the reviewer regarding the revised paper.
>
> ## Remark 1:
> Unclear threat model: can not understand the purposes or capabilities of the threat model.
> ## Answer to remark 1:
> - In the following we explain the threat model and justify its relevance in our case. We added a sentence in the corresponding paragraph of the article to highlight the relevance of the honest-but-curious model in cross-silo FL.
> We consider an honest-but-curious threat model, as described in [1] (which we cite in our paper). In this setting, all participants follow the protocol without any modification, but are free to try to infer information from the intermediate quantities exchanged throughout the protocol.
> -  Such a threat model has been often considered and studied in the literature. In [2], the author stresses the relevance of this model (also called semi-honest): “in many settings, one may assume that although the users may wish to cheat, they actually behave in a semi-honest way” and underlines interest of this semi-honest model from both a theoretical and practical standpoint, given the difficulty to hack code: “In addition to the methodological role of semi-honest parties in our exposition, they do constitute a model of independent interest. In particular, deviation from the specified program, which may be invoked inside a complex software application, is more difficult than merely recording the contents of some communication registers”.
> - In [1], the authors take the example of a Smart grid project where the energy provider has access to the private energy consumption of users. This work states that: “However, a legitimate participant in the protocol such as the energy supplier could not realistically be modeled as a DY [malicious] adversary. In reality, various factors limit the capabilities of the energy supplier including regulations, audits, oversight and desire to maintain reputation.” [...] “We therefore propose to model this agent as a semi-honest] or honest-but-curious (HBC)”
> Insofar as cross-silo FL is often applied in large institutions or companies, in regulated sectors such as banking, insurance or healthcare, the capabilities by the actors are bound in the same fashion as in the smart grid example above. For instance, in the MELLODDY project [3], which is an example of cross-silo FL of interest for this paper in healthcare, each partner is a large company and runs a code that has been previously audited.
> - Therefore, an honest-but-curious behaviour is a realistic model for this setting.
>
> [1]Andrew Paverd, Andrew Martin, and Ian Brown. Modelling and automatically analysing privacy properties for honest-but-curious adversaries. Tech. Rep, 2014.
> [2]  Foundations of cryptography, Oded Goldreich, Volume 2,, page 619, sec 7.2.2
> [3] MELLODDY project, www.melloddy.eu
>
> ## Remark 2:
> Can you please explain the log-likelihood in Eq. 1? I followed reference 50 in the text, but there's no information in the reference.
>
> ## Answer to remark 2:
> We thank the reviewer for this excellent and insightful remark. Indeed, neither reference 50 nor the original Box-Cox paper does provide the derivation of the log-likelihood. In particular the term $(\lambda-1)\mathrm{sign}(x_i) \log (|x_i|+1)$ can be surprising. This term comes from the determinant of the Jacobian of the transformation $\Psi(\lambda, x_i)$. We added in the appendix A.1 that provides the full derivation of the log-likelihood.
>
> ## Remark 3:
> There's no theoretical guarantee of the clients' data privacy leakage from the output of the proposed method: since "the fitted parameter \lambda_* contains all the information revealed during the intermediate steps," whether it can leak any information and whether we should protect it?
>
> ## Answer to remark 3:
> There's no theoretical guarantee of the clients' data privacy leakage from the output of the proposed method.
> This is a limitation of this work, we do not study the privacy leak induced by the final lambda. Our claim is simply that no further information is leaked, as is standard in secure multiparty computation approaches. We added a specific sentence in the limitation paragraph to clarify this point, and leave to future work the study of the amount of information leaked when sharing this parameter.

---

> > ### Author Response · Authors · 2022-08-02
> > **Answer to reviewer W31E Part 2/2**
> >
> > ## Remark 4:
> > The proposed method will destroy the heterogeneity of Federated Learning: After the YJ transformation process, it seems like every client will have the same distribution since they share the same $\lambda_*, \mu_*, \sigma_*$. This poses a risk to FL: if the adversary knows the data distributions of every client (since they have the same distribution), the adversary will quickly deploy the attack of the tails [1]. Moreover, since the threat model is "honest-but-curious", using the YJ transformation will pose a privacy risk to the clients' data since the server can deploy the membership inference attacks [2]. (Also linked to the limitation: The proposed method will destroy the heterogeneity of Federated Learning.)
> > ## Answer to remark 4:
> > - What the reviewer means by “destroying the heterogeneity of Federated Learning” is not clear to us, nor what the benefits of having clients with heterogeneous distributions would be. In particular, most Federated Learning strategies aim to mitigate the detrimental effects of heterogeneity on the efficiency of the training process.
> >
> > - Regarding shared parameters:  sharing the same  $\lambda_*, \mu_*, \sigma_*$ does not imply that every client will have the same distribution, but only that the same transformation will be applied to each client. In particular, applying a common YJ transformation to different distributions leads to different distributions.
> > - Regarding the attack of the tails, first of all, the honest-but-curious setting excludes the possibility of backdoor attacks such as the attack of the tails described in [1].
> > - Besides, applying SecFedYJ neither increases or decreases the possibility of performing the attack of the tails mentioned by the reviewer. This attack of the tails consists in implementing a backdoor in FL on data that are unlikely to appear in the training dataset, as they belong to the tail of the distribution. Indeed, we believe that the attack of the tails is not adapted to our case, as we mainly consider tabular data, whereas this attack is principally designed for images. Besides, the central server only know the marginal distribution columns by column, and not the overall distribution of the dataset.
> > - Regarding the membership attack and taking into account the points raised above, it is unclear to us how knowing the  $\lambda_*, \mu_*$ and $\sigma_*$ parameters would help carrying out this attack, now that we have clarified that clients would not share the same distribution after applying the Yeo-Johnson preprocessing. Could the reviewer please elaborate on this point?
> >
> >
> >
> > ## Remark 5:
> > In Algorithm 3, who will run the algorithm, and what is the job of the clients and the server?
> >
> > ## Answer to remark 5:
> > We thank the reviewer for pointing out that the roles of the different parties in Algo 3 were not detailed enough. We clarified the pseudo-code and added a more detailed explanation of the respective roles of the clients and servers in Appendix D.5. In this algorithm, the server only plays the role of the orchestrator. All the quantities in brackets are shared-secret quantities computed across the client using SMC routines. The new version of the article explicitly details the role of each party.
> >
> > ## Remark 6
> > What is the reason you only consider cross-silo FL since, in my opinion, this method can be generalized to all kinds of FL.
> >
> >
> > ## Answer to remark 6
> > We thank the reviewer for this suggestion.
> > Successfully running a protocol in SMC requires few participants and a stable connection among them. Both of these conditions are met in cross-silo FL. Further, the honest-but-curious model is more adapted to cross-silo, as justified above.
> > In any case, we agree that it would be an interesting direction to investigate whether this method can be adapted to cross-device, especially in the case of unstable connections across clients. We leave this question to future works.
> >
> >
> > ## Remark 7
> > What is the advantage of the proposed method for FL?
> >
> > ## Answer to remark 7
> >  This method ensures that a *common* preprocessing step is performed in each center, which is crucial in e.g. genomics applications in a federated setting. In contrast, performing independent YJ steps in each center would introduce undue heterogeneity, which would be detrimental to downstream tasks as shown by our numerical experiments Figures 5 and 6.

---

> ### Author Response · Authors · 2022-08-08
> **Answer to reviewer W31E**
>
> Dear reviewer W31E,
>
> The discussion deadline is approaching, and we would like to know whether our detailed answer successfully adresses your remarks and questions. If it is the case, we would appreciate if you could reconsider your evaluation. Otherwise, we would be happy to discuss further with you any remaining concerns.
>
> Respectfully,
>
> The authors

---

### Official Review · Reviewer_wXmX · 2022-07-10

**Rating:** 6
**Confidence:** 2
**Soundness:** 4 excellent
**Presentation:** 3 good
**Contribution:** 3 good

**Summary:**

The paper proposes a preprocessing protocol based on multi-party computation that works independently of how the data is skewed among a few parties.


**Questions:**

n/a


**Limitations:**

yes

**Strengths And Weaknesses:**

The algorithm makes clever use of SMPC to minimize cost while preserving security. While it reminds me of secure median computation (Aggarwal et al., J. Cryptol. 2010), it is clearly worthy of independent publication. The interesting aspect is the following: The desired algorithm is iterative. This iteration is replicated in the secure protocol as follows: In every step, there is a relatively simple computation, the result of which (one bit) is revealed. This bit is then used for the next step in the manner of bisectional search. The revealing operation raises the question of the security of doing so. The security is given by the fact that the final result (which is revealed anyway) implies all the intermediate results, and thus, the adversary does not learn any extra information as long as all parties follow the protocol.

The definition of SMC in Section 2 is more limited than the generally used one. The authors only consider computation on local shares but not computation involving communication (line 130). General multi-party computation crucially involves a protocol without which only very limited functionality could be achieved.

The evaluation of the secure computation (Appendix D.3) does not correspond to the state of the art. This might suffice given the low complexity, but I would like to note that there are more efficient ways for secure comparisons (Escudero et al., Crypto 2020).

---

> ### Author Response · Authors · 2022-08-02
> **Answer to reviewer wXmX**
>
> We would like to thank the anonymous reviewer wXmX for their time and positive review. We answer their two main remarks:
>
> - Remark 1: The definition of SMC in Section 2 is more limited than the generally used one. The authors only consider computation on local shares but not computation involving communication (line 130). General multi-party computation crucially involves a protocol without which only very limited functionality could be achieved.
> - Answer: We agree that the example originally provided in this section only considers the case where the SMC routine only involved local operations, while most  non-additive operations require rounds of communication across clients. Indeed, the goal of this section is to introduce the idea of SMC to readers not familiar at all with SMC. We modified the two sentences starting line 130 to provide a more accurate presentation of SMC.
>
> - Remark 2: The evaluation of the secure computation (Appendix D.3) does not correspond to the state of the art. This might suffice given the low complexity, but I would like to note that there are more efficient ways for secure comparisons (Escudero et al., Crypto 2020).
> - Answer: We thank the reviewer for this suggestion, of which we were not aware. We added this reference in Appendix D.3.

---

> > ### Comment · Reviewer_wXmX · 2022-08-08
> > **thanks**
> >
> > I appreciate the author's consideration of my comments.

---

### Author Response · Authors · 2022-08-03
**Answer to the reviewers**

We would like to thank all reviewers for their time and useful remarks. We provided a detailed answer to each reviewer in a separate comment and submitted a revised version of the paper taking into account the reviewers’ remarks. We would be happy to further exchange with the reviewers during the discussion period.

---

> ### Comment · Area_Chair_4HzT · 2022-08-03
> **Acknowledgement**
>
> Dear authors,
> Thank you for your responses and the revision. These will be considered during the discussion.
> Regards,
> Your AC

---

### Comment · Area_Chair_4HzT · 2022-08-05
**Can the round complexity be reduced by parallelism?**

Line 261 says
> Performing SECUREFEDYJ with tmax = 40 takes 726 rounds of communication

Is it possible to reduce the round complexity?

For example, is it possible to partially parallelize the search? If so, what would be the tradeoff in terms of communication per round?

---

> ### Author Response · Authors · 2022-08-05
> **Answer to Area Chair question**
>
> Dear Area Chair,
>
> We would like to thank you for your question and your suggestion. We agree that it would be interesting to see if the clients could simultaneously compute various values of $\Delta$ in SMC for different values of $\lambda$ to speed up the algorithm. However, it would be important to check that no further information is leaked on the data in such a protocol. We did not include such a study in the scope of our work as our work did not focus on the accuracy/speed trade-off (apart from Fig 2 left, see comment below), but we would be happy to mention such an idea in the conclusion of our paper, leaving it to further work.
>
> As we justify in the paper (line 266-270), we think that the overall complexity of our method is compatible with real-world cross-silo FL projects.
>
> Besides, as mentioned line 263 in our work, secFedYJ can be applied in parallel to different features. In other words, only 726 rounds of communication are performed regardless of the dimension of the data.
>
> Moreover, as shown in Fig 2 (left), taking $t_\mathrm{max} = 40$ is quite conservative and can be reduced to decrease the complexity. Indeed, with only $t_\mathrm{max} = 15$, the median relative error between the exponential search and the scikit-learn method based on Brent minimization method is $10^{-4}$.
>
> Finally, the number of communication rounds (726) mentioned is due to the underlying SMC routines used. New SMC protocols are regularly designed and published, which might decrease this number of rounds.
>
> We recall that the exponential search used in secFedYJ has linear convergence while the Brent minimization method traditionally used has a supra-linear convergence. The reason why we choose exponential search instead of Brent minimization method is:
> - It is guarantee to converge to the right value as it leverages the convexity of the negative log-likelihood, while Brent minimization method fails in some cases as shown in Fig 2 (right)
> - It only requires to compute and reveal the sign of the derivative of the log-likelihood, which leak less information. Brent methods requires to compute the exact value of the log-likelihood at each step

---

> > ### Comment · Area_Chair_4HzT · 2022-08-05
> > **Thanks for the response.**
> >
> > Thanks for answering my question.
> >
> > > However, it would be important to check that no further information is leaked on the data in such a protocol.
> >
> > It seems to me that the same argument that is used for exponential/binary search should apply to ternary, quaternary, or k-ary search.

---

> > > ### Author Response · Authors · 2022-08-08
> > > **Answer to AC**
> > >
> > > Dear AC,
> > >
> > > You are totally right. The binary search of secFedYJ can be replaced by a k-ary search. As long as only the sign of the derivative of the log-likelihood is revealed, the proposition 4.1 holds. Indeed, it is straightforward to extend the pseudo-code of $\mathcal{F}$ provided in appendix F of the revised version, for k-ary search.
> > >
> > > We would like to thank you for this suggestion and we would be happy to add the possibility of replacing the binary search to k-ary search in the paper before its publication.

---

### Meta-Review · Area_Chair_4HzT · 2022-08-20

**Recommendation:** Accept
**Confidence:** Less certain

**Metareview:**

This paper studies data preprocesing in the federated learning setting.
It proposes a simple and elegant algorithm for performing a Yeo-Johnson (YJ) power transform on univariate numerical data. This is a nonlinear transform intended to make the data more like a Gaussian.

The paper shows that the likelihood objective is convex and that we can sign its derivative using linear aggregates, which can be performed using secure multiparty computation. This permits us to optimize the transformation using binary search / exponential search. In the honest-but-curious setting, this exponential search does not reveal any superfluous information.

Overall, the reviewers thought this was a valuable contribution and that the paper is well-written and technically sound and that the experimental evaluation is adequate.
However, the reviewers did identify some limitations and directions for further work:
 - *Trust Model.* The paper only considers the honest-but-curious setting. What goes wrong if we consider a more powerful adversary? Can we provide further guarantees?
 - *Leakage via final output.* There is still the possibility of leaking sensitive information via the final parameters $\lambda$, $\mu$, & $\sigma$. This is beyond the scope of MPC guarantees, but it raises the question of whether these techniques can be combined with tools like differential privacy to address this concern.
 - *Efficiency.* The overhead of the system is still quite high in terms of rounds of interaction and total data transfer. The paper appropriately discusses this limitation.
 - *Heterogeneous data.* The paper discusses data heterogeneity, but it is unclear how relevant this is. The algorithm only uses linear aggregates over the coalition, so it shouldn't matter if the data is homogeneous or heterogeneous. Figure 3 compares these two settings, but the results appear to be exactly the same. As such, this figure could be removed from the camera ready version.

On balance, this paper seems appropriate for acceptance at NeurIPS. Data preprocessing is an important task (that is often underappreciated) and the paper proposes and interesting method for doing this in the federated learning setting.

**Award:**

No

---

### Decision · Program_Chairs · 2022-09-14

Accept